# Homozygous receptors for insulin and not IGF-1 accelerate intimal hyperplasia in insulin resistance and diabetes

Qian Li[1,2], Jialin Fu[1,2], Yu Xia[1,2,3], Weier Qi[4], Atsushi Ishikado[1,2], Kyoungmin Park [1,2], Hisashi Yokomizo [1,2], Qian Huang[2], Weikang Cai[2], Christian Rask-Madsen[2], C. Ronald Kahn[2] & George L. King[1,2]*

Insulin and IGF-1 actions in vascular smooth muscle cells (VSMC) are associated with accelerated arterial intima hyperplasia and restenosis after angioplasty, especially in diabetes. To distinguish their relative roles, we delete insulin receptor (SMIRKO) or IGF-1 receptor (SMIGF1RKO) in VSMC and in mice. Here we report that intima hyperplasia is attenuated in SMIRKO mice, but not in SMIGF1RKO mice. In VSMC, deleting IGF1R increases homodimers of IR, enhances insulin binding, stimulates p-Akt and proliferation, but deleting IR decreases responses to insulin and IGF-1. Studies using chimeras of IR(extracellular domain)/IGF1R (intracellular-domain) or IGF1R(extracellular domain)/IR(intracellular-domain) demonstrate homodimer IRα enhances insulin binding and signaling which is inhibited by IGF1Rα. RNA-seq identifies hyaluronan synthase2 as a target of homo-IR, with its expression increases by IR activation in SMIGF1RKO mice and decreases in SMIRKO mice. Enhanced intima hyperplasia in diabetes is mainly due to insulin signaling via homo-IR, associated with increased Has2 expression.

[1] Dianne Nunnally Hoppes Laboratory for Diabetes Complications, Section of Vascular Cell Biology, Joslin Diabetes Center, Harvard Medical School, Boston, MA 02215, USA. [2] Joslin Diabetes Center, Harvard Medical School, Boston, MA 02215, USA. [3] Department of Centre Laboratory, Shandong Provincial Hospital Affiliated to Shandong University, Jinan, Shandong 250001, China. [4] Translational Research and Early Clinical Development, Cardiovascular and Metabolic Research, AstraZeneca, Mölndal, Sweden. *email: George.king@joslin.harvard.edu

Percutaneous coronary angioplasty (PTCA) is an effective treatment for coronary artery diseases[1]. Although drug-eluting stents have significantly reduced the risk of restenosis, it remains a major cause of stent failure[1]. In people with diabetes and insulin resistance, there is a significant increase in the rate of restenosis after PTCA[2], which is related to the increased migration and proliferation of vascular smooth muscle cells (VSMC), as well as influx of inflammatory cells[3,4].

Multiple factors have been suggested to explain the elevated risk of restenosis in association with insulin resistance and diabetes. A long-held assumption is that insulin-like growth factor-1 (IGF-1) and possibly insulin are involved, since both hormones have been reported to stimulate the migration and proliferation of VSMC[5–10]. In addition, circulating insulin and IGF-1 levels are elevated in people with type 2 diabetes[11]. Furthermore, in several rodent models of insulin resistance and type 2 diabetes, intimal hyperplasia after vascular injury is more severe than the non-diabetic or insulin-sensitive controls[12–14]. However, there is a long-standing controversy on whether the accelerated intimal hyperplasia in insulin-resistant and diabetic states are due to the actions of insulin or IGF1, as well as their respective receptors[15].

This lack of clarity on whether insulin or IGF-1 is the main driver of restenosis in diabetes and insulin-resistance states is due to the fact that both insulin and IGF-1 and their receptors have very similar and overlapping cellular and biological responses in VSMC and in many other cell types[16]. The prevailing theory on the actions of insulin and IGF-1 is that insulin binds to insulin receptors (IR)[17,18] at high affinity and mediates mainly the metabolic effects of the hormone, such as those involved on the transport and the metabolism of glucose and amino acids in cells, including VSMC[18–22]. However, IGF-1 and IGF-1 receptors (IGF1R)[23] are involved to a greater extent in the growth-related actions, including cellular migration, DNA synthesis, and possibly extracellular matrix turnover[24–27]. However, it is believed that IGF-1 with minor contribution from insulin enhances the growth activities in VSMC, most likely by binding to IGF-1 receptors. The assumption for a greater role for IGF-1 and IGF-1R are based on the finding that the concentration of insulin to induce VSMC proliferation is 10 to 100 times greater than its physiological levels[6]. Thus, it has been assumed that IGF1R are mainly responsible for the acceleration of arterial intima hyperplasia in diabetes and insulin-resistant states.

In order to determine the relative importance of IR and IGF1R in mediating the growth of VSMC to accelerate intimal hyperplasia in vivo, we have deleted IR or IGF1R from VSMC and determined their actions on VSMC proliferation and intimal hyperplasia both in culture and in vivo. The mechanism of insulin and IR's growth-promoting effects compared with IGF1R have been also characterized by analysis of IR and IGF1R structures with identification of IR's downstream target in VSMC by RNA sequencing. Surprisingly, these studies reveal that insulin acting via IR is mainly responsible for enhancing VSMC's actions to accelerate intimal hyperplasia in insulin resistance and diabetes and that this occurs by its effects on gene expression, involved in the extracellular matrix remodeling, like hyaluronan synthase.

## Results

### Characterization of IR deletion in VSMC of *SMIRKO* mice.
To characterize the function of IR in VSMCs in vivo, IR were deleted in VSMC by breeding *IR flox/flox* mice with *SM22α –Cre* transgenic mice. These *SMIRKO* mice had no detectable IR in the aorta. In contrast, the expressions of IR in the brain, liver, kidney, and adipocyte were similar between *SMIRKO* and *WT* mice, although IR were partially reduced in the heart of *SMIRKO* mice,

as expected (Supplementary Fig. 1). Insulin stimulation of Akt phosphorylation in vivo revealed an $8.0 \pm 1.8$-fold increase in p-Akt in the aorta of *WT* mice, but only $2.7 \pm 0.8$-fold in *SMIRKO* mice (Supplementary Fig. 2). By contrast, insulin-induced p-Akt in the liver and skeletal muscle were similar in the two groups of mice (Supplementary Fig. 2). Although IR expressions were reduced partially in the myocardium of *SMIRKO* vs. *WT* mice, insulin's induction of p-Akt in the heart was similar (Supplementary Fig. 2). VSMC from *WT* and *SMIRKO* mice also exhibited selective deletion of IR in VSMC from *SMIRKO*, but IR levels in the endothelial cell from *SMIRKO* and *WT* mice were not different (Supplementary Fig. 1D).

### Intimal hyperplasia of femoral artery induced by wire injury.
The extent of intimal hyperplasia of the femoral artery after wire injury was determined in *WT* and *SMIRKO* mice, placed on a HFD for 8 weeks. Body weight, blood pressure, plasma insulin, lipid levels, intraperitoneal glucose tolerance test (IPGTT), and insulin tolerance tests (IPITT) did not differ between the two groups of mice (Supplementary Fig. 3).

The extent of intimal hyperplasia assessed by elastin staining in the femoral artery of *SMIRKO* mice was decreased by 37% compared with those of *WT* mice (Supplementary Fig. 4A, B). The ratio of intima/media of *SMIRKO* mice was also decreased compared with *WT* mice (Supplementary Fig. 4C), although the media area was not different (Supplementary Fig. 4D). VSMC proliferation, as determined by BrdU and SM22a double staining at 7 days after injury was decreased by 51% in *SMIRKO* mice compared with *WT* mice (Supplementary Fig. 4E, F). Proliferation of VSMC in the femoral artery after wire injury as assessed by *cyclin A2* mRNA expression increased 3.6-fold in WT mice, which was significantly decreased by 32% in *SMIRKO* mice (Supplementary Fig. 4G). Similarly, cellular proliferation of VSMCs cultured from aorta showed insulin and IGF-1 increased Edu incorporation by 92 and 232%, respectively in *WT* VSMCs, but only by 41 and 143% in *SMIRKO* mice, respectively (Supplementary Fig. 4H). IGF1, but not insulin, significantly increased proliferation of VSMCs from *SMIRKO* mice.

To exclude the effect of IR deletion in the heart and macrophages, IR were also knocked out by Cre recombinase driven by *Myh11* promoter (*Myh11IRKO*). IR expression was significantly decreased in the aorta of Myh11IRKO mice compared with WT mice, whereas its expressions were not different in the heart, skeletal muscle, and liver between two groups of mice (Supplementary Fig. 5). When the mice were subjected to femoral artery injury, the intimal area and intima/media ratio were decreased significantly by 49 and 60%, respectively, in the femoral artery of *Myh11dIRKO* mice. The thickness of arterial media was not different between two groups of mice (Fig. 1a–d). VSMC proliferation, as determined by BrdU and SM22a double staining at 7 days after injury, was significantly decreased in *Myh11IRKO* mice compared with *WT* mice (Fig. 1e, f). Cellular proliferation of cultured aortic VSMCs, determined by EdU incorporation, increased by 2.7-fold with insulin in *WT* VSMCs, but only 1.5-fold in *Myh11IRKO* VSMCs (Fig. 1g).

### Deletion of IGF1R in VSMC of *SMIGF1RKO* mice.
*SMIGF1RKO* mice were generated by breeding *IGF1R flox/flox* mice with Myh11-Cre transgenic mice. IGF1R protein expression was decreased by 70% in the aorta of *SMIGF1RKO* mice compared with *WT* mice, which was unchanged in the heart and brain (Supplementary Fig. 6). The selectivity of IGF1R deletion in VSMC was further confirmed since no differences in IGF1R protein expressions were observed in endothelial cells from *SMIGF1RKO* vs. *WT* mice (Supplementary Fig. 6). After HFD,

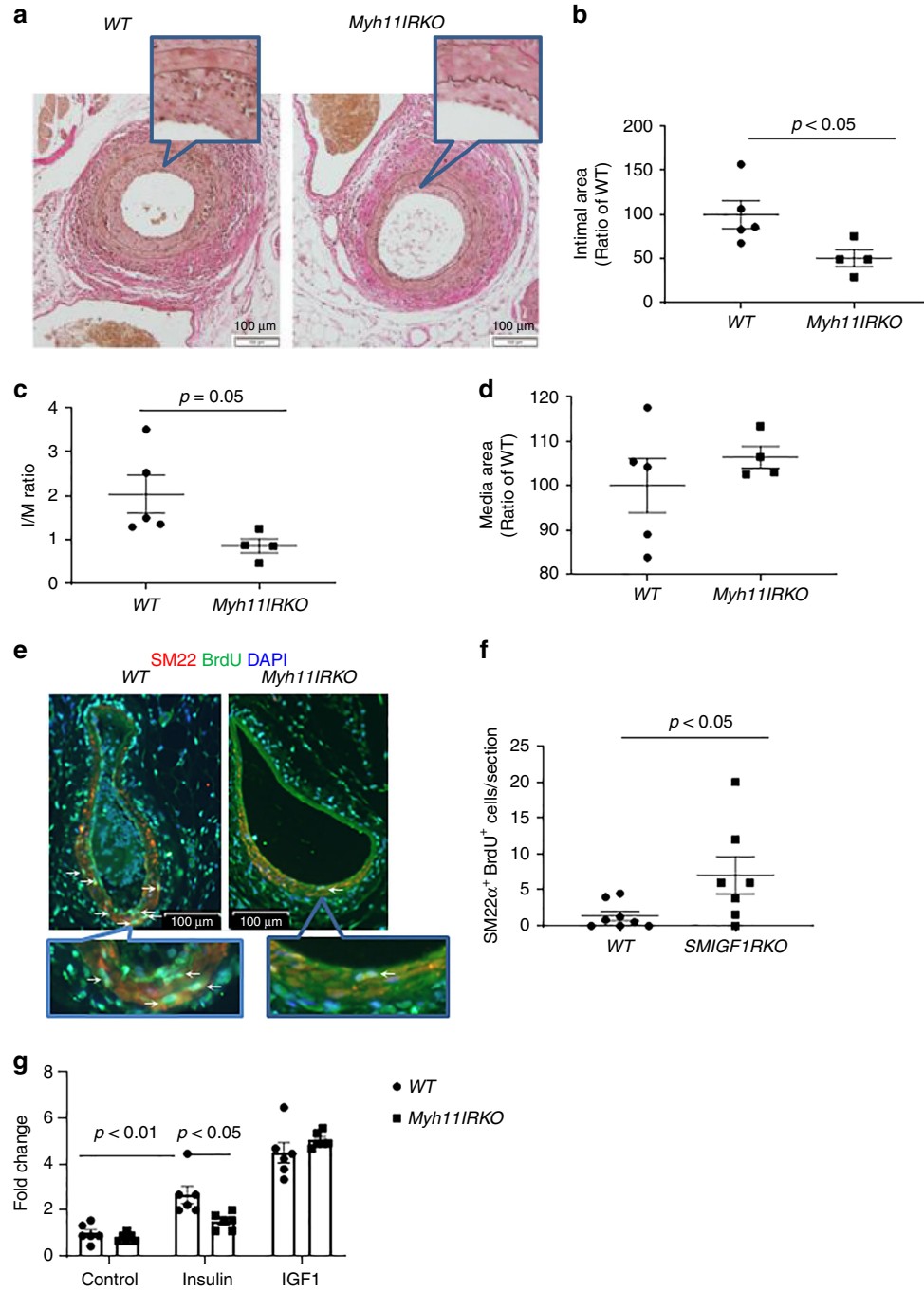

**Fig. 1** Wire injury-induced femoral artery intimal hyperplasia in HFD-fed *WT* and *Myh11IRKO* mice. **a**–**d** Intimal hyperplasia was determined by elastin staining. **a** indicates representative images. Summarized intimal area (**b**), intima/media (I/M) ratio (**c**), and media area (**d**) are shown (*WT*, $n = 5$; *Myh11IRKO*, $n = 4$). **e**, **f** VSMC proliferation in the wire-injured femoral artery. The femoral arteries were subjected to wire injury, and then the mice were infused with Brdu with osmotic minipump for 1 week. VSMCs proliferation was determined by SM22α and Brdu double staining. **e** indicats representative images, and **f** shows summarized data (*WT*, $n = 8$; *Myh11IRKO*, $n = 7$). **g** Insulin and IGF-1 induced VSMCs proliferation. Starved *WT* or *Myh11IRKO* VSMCs were stimulated with insulin (10 nM) or IGF-1 (10 nM) for 24 h, and cell proliferation was determined by Edu incorporation and measured by flow cytometry ($n = 6$ per group). The data are mean ± SEM. Two-tailed *t* test (**b**, **c**, **d**, **f**) or two-way ANOVA with a post hoc test (**g**). Source data are provided as a Source Data file

body weights, plasma lipid, IPGTT, and IPITT were not different between the two groups of mice (Supplementary Fig. 7A–E). Intimal area of the femoral artery was increased by 45% in *SMIGF1RKO* mice compared with *WT* mice (Fig. 2a, b). The intima/media ratio, which is the gold standard for restenosis, and media area were not different between *WT* and *SMIGF1RKO* mice (Fig. 2c, d).

Sections of wire-injured femoral artery from *WT* or *SMIGF1RKO* mice were stained with VSMC marker SM22α or macrophages marker Mac2. VSMC numbers in the intimal area of femoral artery from *SMIGF1RKO* mice were increased by 50% compared with that of *WT* mice. The macrophages and the collagen content (trichrome staining) in the intimal area of femoral artery of *WT* mice were not different compared with that

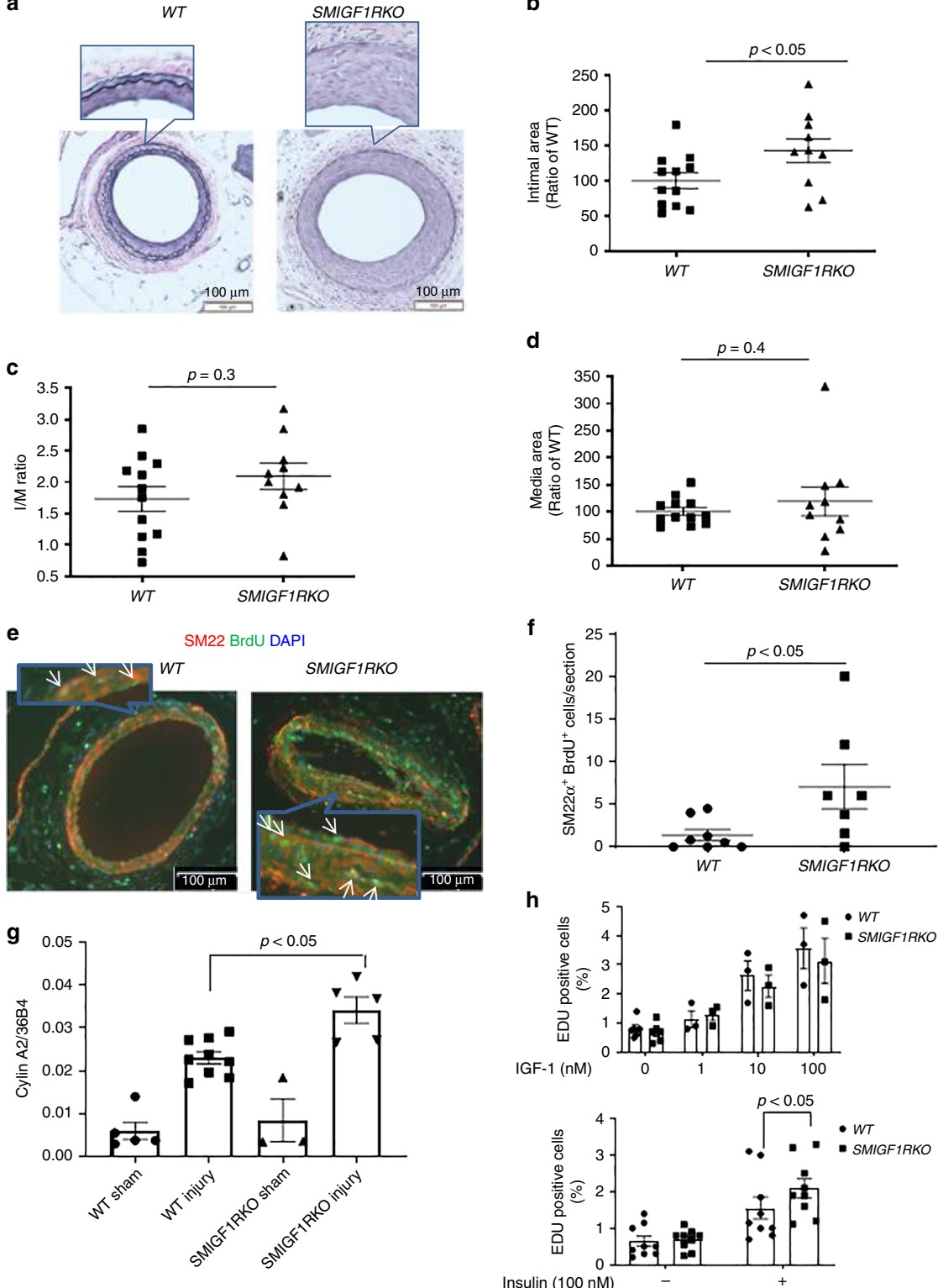

**Fig. 2** Wire injury-induced femoral artery intimal hyperplasia in HFD-fed *WT* and *SMIGF1RKO* mice. **a–d** Intimal hyperplasia of femoral artery was determined by elastin staining at 4 weeks after wire injury. **a** indicates representative images. Summarized intimal area (**b**), intima/media (I/M) ratio (**c**), and media area (**d**) are shown (*WT*, n = 12; *SMIGF1RKO*, n = 10). E-F VSMCs proliferation. The femoral arteries were subjected to wire injury, and then the mice were infused with Brdu with osmotic minipump for 1 week. VSMCs proliferation was determined by SM22α and Brdu double staining. **e** indicates representative images, and **f** shows summarized data (*WT*, n = 8; *SMIGF1RKO*, n = 7). **g** *Cyclin A2* gene expression. *Cyclin A2* gene expression in femoral artery at 9 day after wire injury was used as a surrogate marker of cell proliferation, and was determined by qPCR (*WT* sham, n = 5; *WT* injury, n = 9; *SMIGF1RKO* sham, n = 3; *SMIGF1RKO* injury, n = 5). **h** IGF-1 and insulin-induced *WT* or *SMIGF1RKO* VSMCs proliferation. Starved VSMCs were stimulated with 1–100 nM IGF-1 or 100 nM insulin (n = 9 for each group) for 24 h, and cell proliferation was determined by Edu incorporation and measured by flow cytometry. All data from this figure are mean ± SEM. Two-tailed *t* test (**b**, **c**, **d**, **f**) or two-way ANOVA with a post hoc test (**g**, **h**). Source data are provided as a Source Data file

of *SMIGF1RKO* mice (Supplementary Fig. 8). Proliferation of VSMC, as measured by BrdU and SM22 double staining, was increased by fivefold in wire-injured femoral artery of *SMIGF1RKO* vs. *WT* mice (Fig. 2e, f), which was confirmed by increase (48%) of *cyclinA2* mRNA expression in the femoral artery of *SMIGF1RKO* compared with that of *WT* mice (Fig. 2g). Proliferation of cultured VSMC, as evaluated by Edu incorporation, showed that the effect of IGF-1 (1 to 100 nM) was similar in VSMC from the two types of mice (Fig. 2h). In contrast, VSMC proliferation induced by 100 nM insulin was increased by 35% in cells from *SMIGF1RKO* mice compared with *WT* mice (Fig. 2i).

**Insulin and IGF1 signaling in VSMCs.** To investigate the mechanisms underlying the findings of decreased intimal hyperplasia in *SMIRKO* mice, but not in *SMIGF1RKO* mice, insulin and IGF-1's signaling in cultured VSMCs from *SMIRKO*, *SMIGF1RKO*, and *WT* mice were studied. In cells from the *SMIRKO* mice, IR content was decreased > 95%, without changes in IGFIR protein expression. Conversely, in VSMCs from *SMIGFIRKO* mice, IGF1R expression was decreased by 70% without changes in IR expressions (Fig. 3a, b).Tyrosine phosphorylations at 1150/1151 of IR (pTyr-IR) and 1135/1136 of IGF1R (pTyr IGF-IR) can be assessed using the same antibody, but the receptors can be differentiated by molecular weights, since IGF1Rβ is 2 kDa greater than IRβ[28,29] (Fig. 3a). Insulin increased pTyr-IR significantly at Y1150/1151 in *WT* VSMCs in a dose-dependent manner reaching maximum at 100 nM. No pTyr-IR was observed in *SMIRKO* VSMCs even at 100 nM insulin, although insulin increased pTyr-IGF1R (Fig. 3a). Unexpectedly, insulin-induced pTyr-IR was significantly increased to greater levels in *SMIGF1RKO* compared with *WT* VSMCs. Insulin stimulated pTyr-IR in a dose-dependent manner from 1 nM to 100 nM in both *WT* and *SMIGF1RKO* mice, but the effects of insulin were more pronounced in *SMIGF1RKO* mice than *WT* mice. Levels of pTyr-IR were increased fivefold greater by insulin at 100 nM in *SMIGF1RKO* VSMCs compared with *WT* VSMCs (Fig. 3a). In parallel with the changes in pTyr-IR, insulin increased p-Akt in a dose-dependent manner in *WT* VSMCs, and its effects were attenuated in *SMIRKO* mice by more than 60%. Consistent with the elevated levels of pTyr-IR in *SMIGF1RKO* VSMCs, insulin-stimulated p-Akt was also significantly greater than that in *WT* VSMC after stimulation with 10 nM or 100 nM insulin (Fig. 3a). Insulin also induced p-Erk in *WT* VSMCs from 1 nM to 100 nM, which was significantly lower in *SMIRKO* compared with *WT* VSMC (Fig. 3a). Interestingly, insulin-induced p-Erk was not enhanced in *SMIGF1RKO* VSMC compared with *SMIRKO* VSMC. Furthermore, PDGF-induced p-Akt and p-Erk were comparable between *WT* and *SMIRKO* VSMCs (Supplementary Fig. 9).

IGF-1-induced pTyr-IGF1R levels were decreased by 26 and 50% in *SMIRKO* VSMCs and in *SMIGF1RKO* VSMCs, respectively, compared with *WT* VSMCs (Fig. 3b). Similarly, IGF-1-induced p-Akt in *WT* VSMCs from 1–100 nM by 2.7 to 16.5-fold, but IGF-1-induced p-Akt was lower in *SMIGF1RKO* VSMCs compared with *WT* VSMCs (Fig. 3b). IGF-1 (100 nM)-induced p-Erk was also decreased by 39 and 45%, respectively, in *SMIRKO* VSMCs and in *SMIGF1RKO* VSMCs as compared with *WT* VSMCs (Fig. 3b).

**Characterization of homo-IR/IGF1R and hetero-IR/IGF1R on signaling.** To determine the mechanisms for the enhancement of insulin signaling, we examined insulin and IGF1 binding in *SMIRKO* and *SMIGF1RKO* VSMCs. Compared with VSMCs from *WT* mice, insulin binding was decreased by >80% in *SMIRKO* VSMCs (Fig. 4a). In contrast, $^{125}$I-insulin binding was

increased by twofold in *SMIGF1RKO* VSMCs compared with *WT* VSMCs (Fig. 4a). $^{125}$I-IGF1 binding was decreased by 50% in *SMIGF1RKO* VSMCs and by 20% in *SMIRKO* VSMCs compared with *WT* VSMC (Fig. 4b). To investigate whether the increased insulin binding to *SMIGF1RKO* VSMCs was due to increases of IR on the cell surface, cell surface proteins were labeled with biotin which were precipitated with avidin-conjugated beads and analyzed by immunoblotting. IR protein expressions on cell surface or in the whole cell were not different between *WT* and *SMIGF1RKO* VSMCs (Fig. 4c, d). Protein levels of IGF1R were decreased on the cell surface of *SMIGF1RKO* VSMCs compared with *WT* VSMCs by 63%, and not changed in *SMIRKO* vs. *WT* VSMCs (Fig. 4c, d).

Structurally, IR and IGF1R are composed of heterodimers of α and β chains conjugated by disulfide bonds[16]. In addition, IR and IGF1R can form hetero-IR/IGF1R receptors due to their structure similarities[30]. To estimate the relative levels of homo-IR, hetero-IR/IGF, and homo-IGF1R receptors, cell lysates were immunoprecipitated with anti-IGF1R antibody, and the levels of IR and IGF1R in immunoprecipitation, supernatant, and cell lysate before immunoprecipitation were determined. The IR in immunoprecipitation represented IR from hetero-IR/IGF1R, whereas IR remained in supernatant were likely homo-IR/IR. IGF1R was undectable in supernatant in all three groups of cells, indicating that the difference of IR in supernatant of *WT* and *SMIGF1RKO* was not due to incomplete precipitation. The results showed that ~20% of IR was homo-IR receptors in *WT* VSMCs compared with 40% in *SMIGF1RKO* VSMCs, which were due to a 70% reduction of IGF1R expression (Fig. 4e, f).

To compare clearly homo-IR and homo-IGF1R signaling and activation without the presence of IR/IGF1R heterodimers, VSMCs from *IR flox/flox* mice were infected with AdCre to knockdown IR. IR was deficient in AdCre transfected VSMCs. Insulin-induced Akt phosphorylation was increased by 6.4-fold in *WT* VSMCs, but only threefold in IR-deficient VSMCs (Supplementary Fig. 10A). *IGF1R flox/flox* VSMCs were infected with AdCre to completely delete IGF1R. Infection with AdCre-deleted IGF1R in *IGF1R flox/flox* VSMCs by >99% (Supplementary Fig. 10B). Consistent with the previous findings from VSMC of *SMIGF1RKO* mice, insulin (100 nM)-induced p-Akt was enhanced significantly by 340%, whereas IGF-1-induced p-Akt was decreased by 63% in VSMCs with IGF1R knockdown compared with *WT* VSMCs (Supplementary Fig. 10B). These results clearly demonstrate that homo-IR can transmit insulin signaling greater than hetero-IR/IGFIR by 340% and homo-IGF1R.

To clarify further the mechanism for the enhancement of homo-IR signaling compared with hetero- IR/IGF1R or homo-IGF1R, we characterized whether the increased activities were located in the IR and IGF1Rα subunit containing the ligand-binding motif or their β subunit, which has the tyrosine kinase[16]. Ligand binding to the α subunit leads to β subunits trans-autophosphorylation[16]. There are two possibilities for the decrease in the activation of hetero-IR/IGF1R compared with the homo-IR by insulin. The first possibility is that insulin bound less to IRα/IGF1α than IRα/IRα. The second possibility is that insulin has similar binding affinity to homo-IR and hetero-IRα/IGF1Rα, but insulin binds to IR/IGF1R is not as effective to activate IRβ/IGF1Rβ. To differentiate these possibilities, we studied the effects of insulin and IGF-1 to activate specifically constructed chimeric receptors which are composed of IR-extracellular domain (ECD)/IGF1R-intracellular domain (ICD) or IGF1R-extracellular domain (ECD)/IR-intracellular domain (ICD). First, both IR and IGF1R were deleted completely by infecting VSMCs containing both *IR flox/flox* and *IGF1R flox/flox* with AdCre. In those double-knockout (DKO) VSMCs, with both

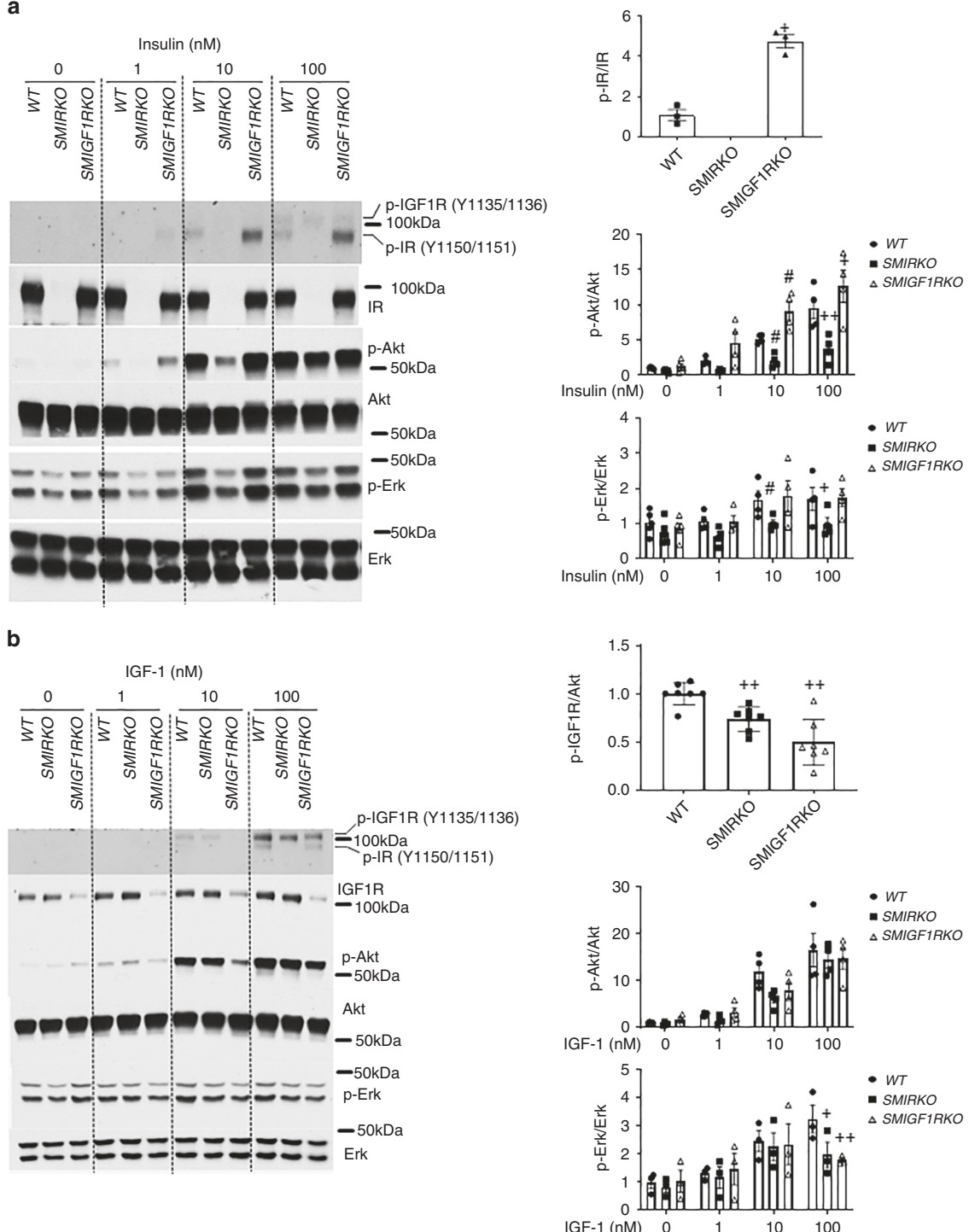

**Fig. 3** Insulin and IGF-1 signaling in *WT*, *SMIRKO*, or *SMIGF1RKO* VSMCs. **a** Starved VSMCs were stimulated with different dose of insulin for 15 min (For pIR/IR, $n = 3$ for each group; for pAkt/Akt and pErk/Erk, $n = 5$ for each group at 0 nM; $n = 4$ for each group at 1, 10, and 100 nM. $^{#}p < 0.05$, vs. WT stimulated with 10 nM insulin; $^{+}p < 0.05$, $^{++}p < 0.01$ vs. *WT* stimulated with 100 nM insulin). **b** Starved VSMCs were stimulated with different dose of IGF-1 for 15 min (For pIR/IR, $n = 7$ for each group; for pAkt/Akt, $n = 4$ for each group and for pErk/Erk, $n = 3$ for each group, $^{+}p < 0.05$, $^{++}p < 0.01$ vs. *WT* stimulated with 100 nM IGF-1). The data are mean ± SEM. Two-tailed *t* test or two-way ANOVA with a post hoc test. Source data are provided as a Source Data file

IR and IGF1R deleted, either chimeric receptor IR-ECD/IGF1R-ICD or IGF1R-ECD/IR-ICD was overexpressed using adenovirus vectors to mediate the infection. Both endogenous IGF1Rβ and IRβ expressions were decreased by >90% in VSMC with double knockdown of IR and IGF1R (Fig. 5a, b). In VSMCs over-expressed with IR-ECD/IGF1R-ICD, the amount of the IGF1Rβ was increased by 1.7-fold compared with endogenous IGF1Rβ in WT VSMCs. IRβ was increased by 13.7-fold in VSMCs overexpressing the chimeric IGF1R-ECD/IR-ICD compared with endogenous IRβ in WT VSMCs. Insulin-induced p-Akt levels were greater in IR-extracellular domain/IGF1R-intracellular domain VSMCs by 250% than IGF1R-extracellular domain/

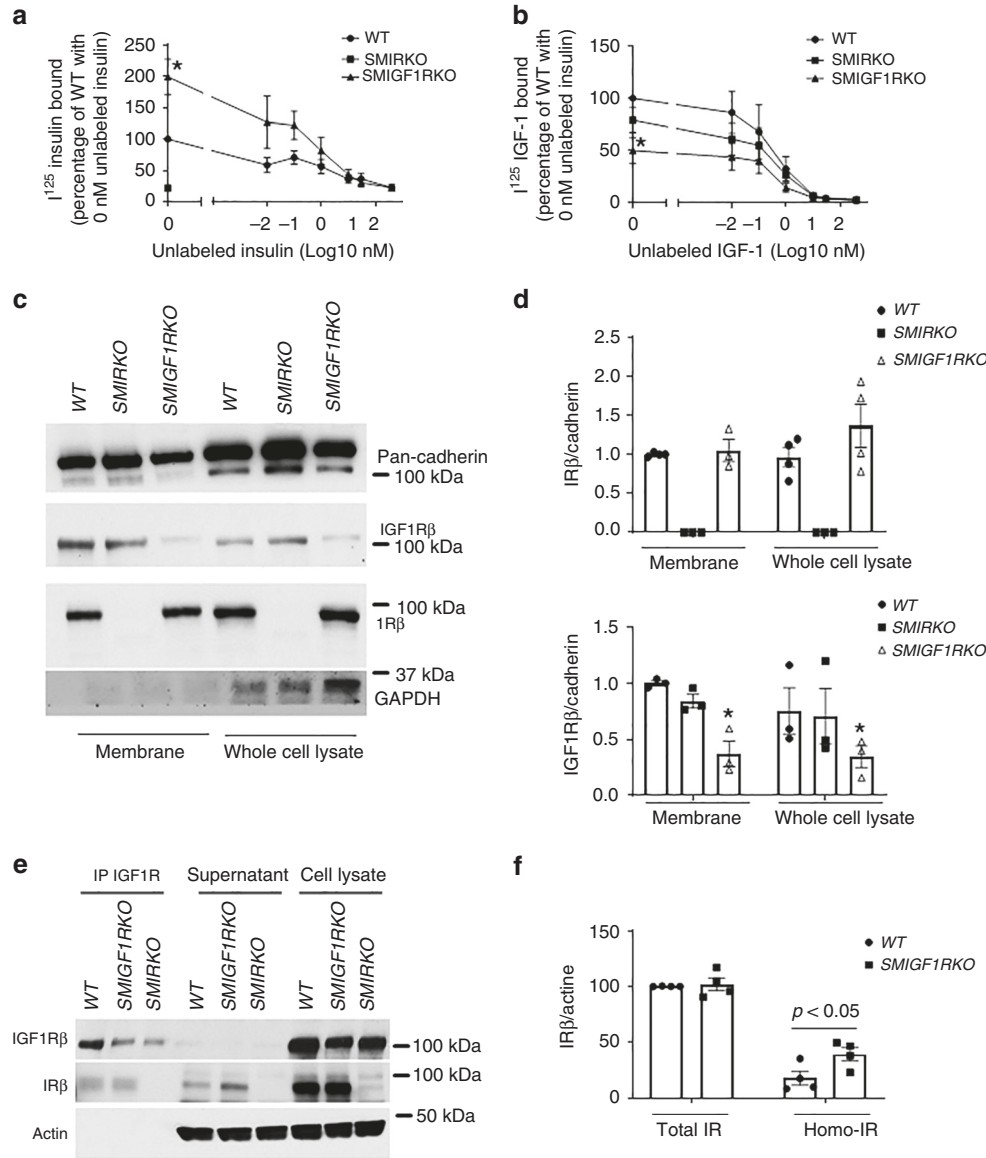

**Fig. 4** Characterization of homo-IR/IGF1R and hetero-IR/IGF1R on signaling in *WT, SMIRKO,* or *SMIGF1RKO* VSMCs. **a, b** Insulin (**a**) or IGF-1 (**b**) binding. The VSMCs were incubated with mixture of $I^{125}$-labeled insulin and different dose of unlabeled insulin (**a**) or mixture of $I^{125}$-labeled IGF-1 and different dose of unlabeled IGF-1 (**b**) for 5 h on ice. Insulin or IGF-1 binding was determined by scintillation counter (**a**, $n = 5$ for *WT* and *SMIGF1RKO* at 0 nM unlabeled insulin; $n = 3$ for *WT* and *SMIGF1RKO* at 0.01, 30, and 400 nM unlabeled insulin; $n = 4$ for *WT* and *SMIGF1RKO* at 0.1, 1, and 10 nM unlabeled insulin; **b**, $n = 3$ for each group; *$p < 0.05$ *SMIGF1RKO* vs. *WT*). **c, d** IR and IGF1R on cell surface. The cell membrane protein was labeled with biotin. The biotin-conjugated protein was precipitated by avidin-conjugated beads. The levels of IGF1R and IR on cell surface were determined by western blotting ($n = 4$ for *WT* memtrane, *WT* total, and *SMIGF1RKO* total; $n = 3$ for SMIRKO membrane, SMIRKO total, and SMIGF1RKO membrane, *$p < 0.05$ *SMIGF1RKO* vs. *WT*). **e, f** Ratio of homo-IR and hetero-IR/IGF1R. The cell lysates were immunoprecipitated with anti-IGF1R antibody. The levels of IR and IGF1R in immunoprecipitation, supernatant after immunoprecipitation with anti-IGF1R antibody and whole-cell lysates without immunoprecipitation were determined by western blotting. The IR in immunoprecipitation was IR from hetero-IR/IGF1R, and the remaining IR in supernatant was homo-IR. The IR in whole-cell lysates included homo-IR and IR in hetero-IR/IGF1R ($n = 4$ for each group). The data are mean ± SEM. Two-tailed *t* test. Source data are provided as a Source Data file

IR-intracellular domain VSMCs, whereas p-Akt induced by IGF-1 in IGF1R-ECD/IR-ICD VSMCs and IR-ECD/IGF1R-ICD VSMCs were comparable (Fig. 5a–d). These results support strongly the conclusion that the enhanced effect of insulin in SMIGF1RKO vs. WT were due to enhanced binding of insulin to the IRα /IRα homodimers, and not due to activation of IRβ/IRβ homodimers. To support further that IR/IR homodimers are responsible for enhancing insulin's actions in VSMCs from SMIGF1RKO mice, VSMCs with complete deletions of both endogenous IR and IGF1R were overexpressed with IR-ECD/IGF1R-ICD alone, IGF1R-ECD/IR-ICD alone or both IR-ECD/

IGF1R-ICD and IGF1R-ECD/IR-ICD. The results showed that insulin's activation of p-Akt in cells overexpressing IR-extracellular domain/IGF1R-intracellular domain was sevenfold greater than WT cells and IGF1R-extracellular domain/IR-intracellular domain cells as before. In addition, insulin's effect on p-Akt in cells expressing IR-ECD/IGF1R-ICD and IGF1R-ECD/IR-ICD together were greatly decreased compared with cells which expressed only IR-ECD/IGF1R-ICD (Fig. 5e, f).

**Insulin targets in VSMCs.** To identify the specific downstream cellular targets of IR that are mediating its enhanced effects on

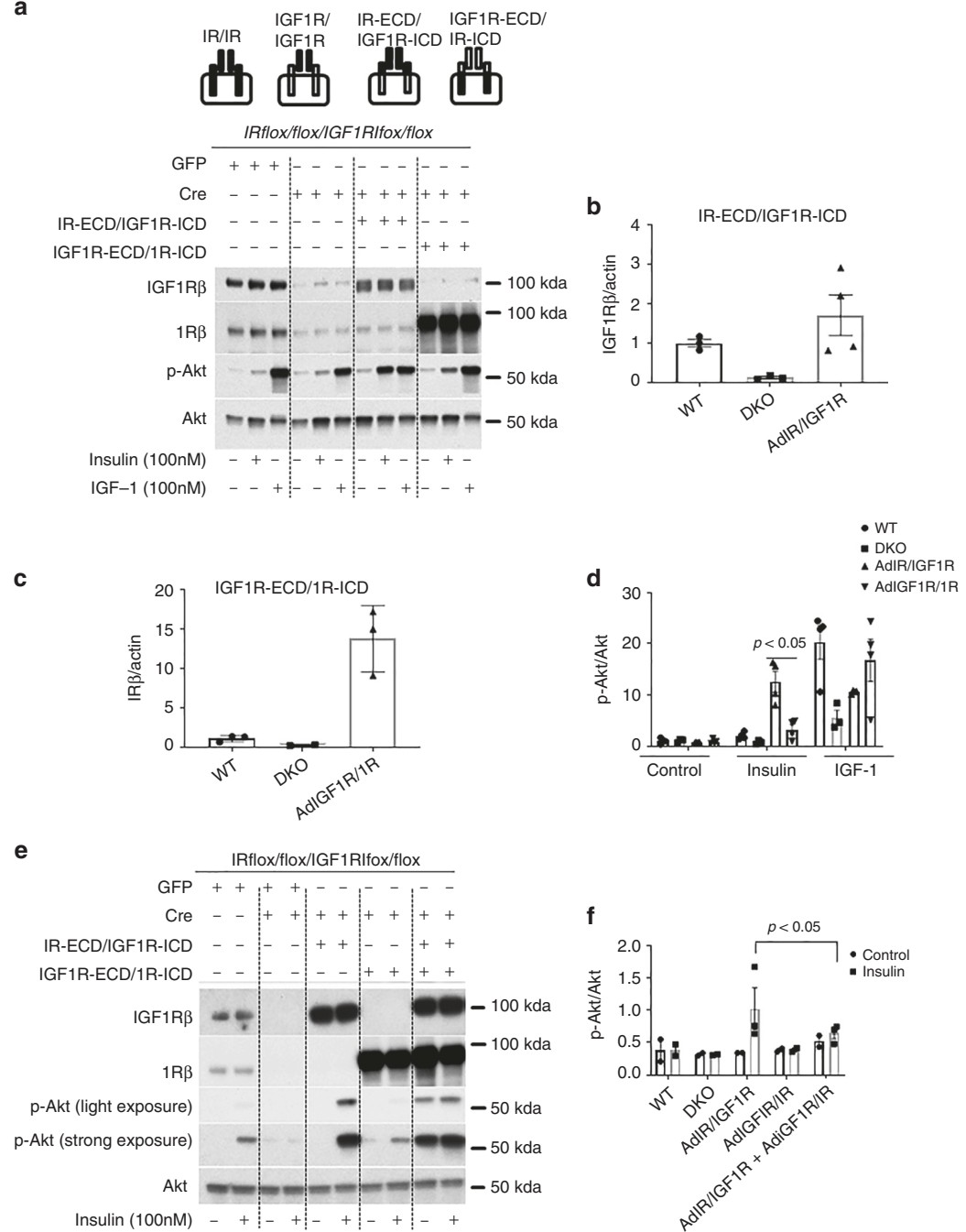

**Fig. 5** Insulin and IGF-1 action on chimeric IR/IGF1R receptors. VSMCs were cultured from aortas of mice which had both *IRflox/flox* and *IGF1Rflox/flox* alleles. Then IR[flox/flox]IGF1R[flox/flox] VSMCs were infected with AdCre to knockdown both IR and IGF1R. The IR and IGF1R double knockdown VSMCs were further infected with AdIR-ECD/IGF1R-ICD or AdIGF1R-ECD/IR-ICD or both to overexpress chimeric IR-ECD/IGF1R-ICD or IGF1R-ECD/IR-ICD receptors. **a**, **e** indicate representative western blotting images. **b**–**d**, **f** show summarized quantification data (**b**, $n = 3$ for WT and DKO, $n = 4$ for AdIR/IGF1R; **c**, $n = 3$ for each group; **d**, $n = 4$ for WT, AdIR/IGF1R, and AdIGF1R/IR, $n = 3$ for DKO; **f**, $n = 3$ for each group). The data are mean ± SEM. Two-way ANOVA with a post hoc test. Source data are provided as a Source Data file

VSMC proliferation and intimal hyperplasia, cultured VSMCs with IGF1R deletion of >99% derived by infecting VSMCs from *SMIGFIRKO* mice with AdCre and VSMC from *WT* mice were stimulated with insulin at 100 nM, and changes in gene expression were evaluated by RNA-seq analysis (data shown in Supplementary Table 1). Insulin increased the expression of 13 genes by >1.5-fold in *IGF1R KO* VSMC than *WT* VSMC. These genes were evaluated further in the media of the aorta from *WT*, *SMIRKO*, and *SM1GF1KO* as shown in Fig. 6. Of these 13 genes,

only hyaluronan synthesis 2 (*Has2*) was decreased in the aorta from *SMIRKO* mice compared with *WT* mice by >50% ($p < 0.05$) (Fig. 6a, b). Furthermore, *Has2* mRNA and protein expressions were increased in the aorta of *SMIGF1RKO* mice compared with *WT* mice (Fig. 6c). These results suggest that changes in *Has2* could be a specific cellular target of IR in VSMC to enhance DNA synthesis. To evaluate the regulation of *Has2* expression by insulin, cultured VSMC from *WT* and *SMIGF1R KO* mice were compared. Insulin (10 nM) significantly increased *Has2* mRNA

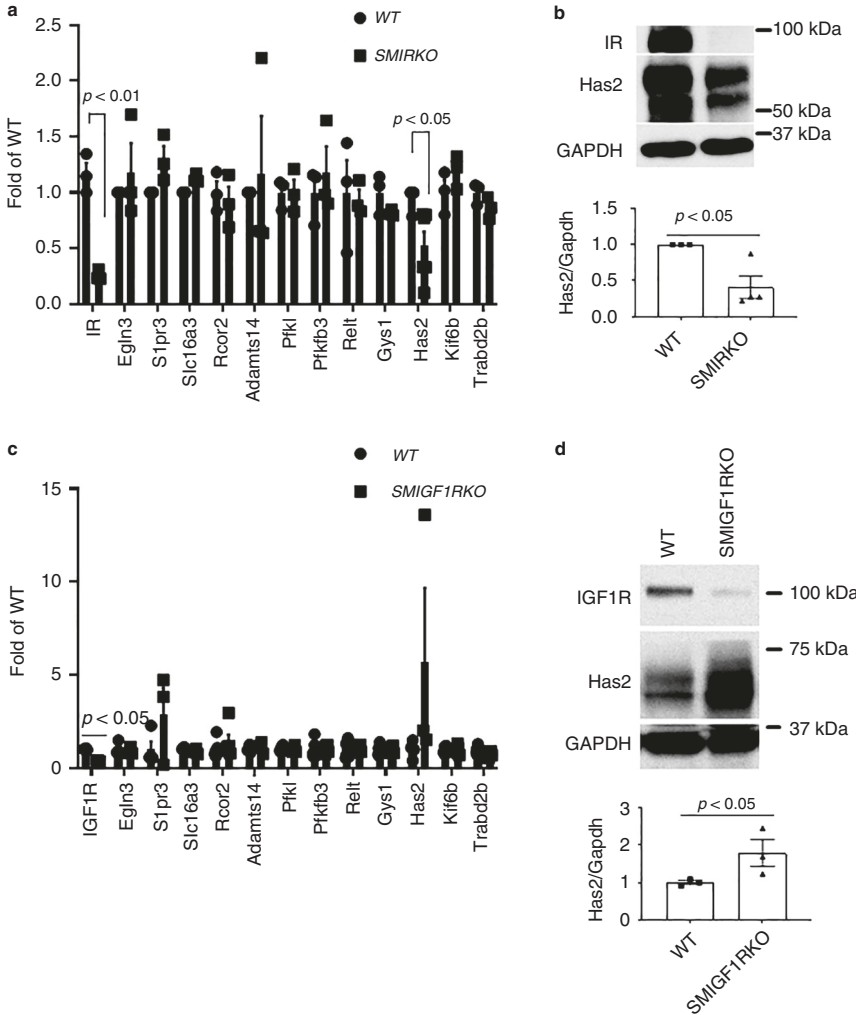

**Fig. 6** Insulin targets in VSMCs. **a** The expression of insulin-targeted genes in aorta media of *WT* and *SMIRKO* mice. RNA was extracted from aorta media of *WT* and *SMIRKO* mice and the expression levels of genes were determined by qPCR (*WT* Has2, $n = 5$; *SMIRKO* Has2, $n = 6$; $n = 3$ for each of the remaining groups). **b** The expression of Has2 protein in aorta media of *WT* and *SMIRKO* mice (*WT*, $n = 3$; *SMIRKO*, $n = 4$). **c** The expression of insulin-targeted genes in aorta media of *WT* and *SMIGF1RKO* mice. RNA was extracted from aorta media of *WT* and *SMIGF1RKO* mice, and the expression levels of genes were determined by qPCR ($n = 3$ per group). **d** The expression of Has2 protein in aorta media of *WT* and *SMIGF1RKO* mice ($n = 3$ per group). The data are mean ± SEM. Two-tailed t test. Source data are provided as a Source Data file

levels at 1 h by 248% in *SMIGF1RKO* VSMCs (Fig. 7a, b). In addition, insulin also increased hyaluronan content in the culture media of *WT* VSMC, whereas *SMIRKO* VSMC did not respond to insulin. In contrast, insulin significantly increased hyaluronan content in the medium of *SMIGF1RKO* VSMC (Fig. 7c).

The expression of *Has2* gene was further investigated in vivo using femoral artery 9 days after wire injury in *WT* (*IR flox/flox*), *SMIRKO*, *WT* (*IGF1R flox/flox*), and *SMIGF1RKO* mice on HFD diet (Fig. 7d). The expression of *Has2* mRNA was decreased in femoral artery of *SMIRKO* mice compared with its *WT* control (*IR flox/flox mice*). In contrast, the mRNA expression of *Has2* in the femoral artery was significantly increased in *SMIGF1RKO* mice vs. its *WT* control, *IGF1R flox/flox* mice. The intensity of hyaluronan staining as levels in the intimal area of femoral artery of *SMIGF1RKO* mice were significantly increased compared with *WT* (*IGF1R flox/flox*), as well as to *IR flox/flox* and *SMIRKO* mice (Fig. 7e, f).

Analysis of insulin signaling pathways, which could be responsible for regulating *Has2* expression, was also studied using VSMC from *SMIGF1RKO* and *WT* mice. Again, insulin increased *Has2* mRNA levels greater in *SMIGF1RKO* VSMC than

*WT* VSMC (Supplementary Fig. 11). Small-molecule inhibitors of PI3 Kinase (wortmannin) and MEK (PD98059) inhibited completely the expression of insulin-induced *Has2* expression (Supplementary Fig. 11).

FoxO and mTOR are the two major downstream targets of Akt. Insulin increased phosphorylation of FoxO1 at threonine 24 and FoxO3a at threonine 32 (Supplementary Fig. 12). Recently, it was reported that FoxO1 could bind to the promoter of *Has2* and inhibit the transcription of *Has2*[31]. Thus, we assessed the effect of insulin to regulate *Has2* gene expression by infecting VSMCs with adenovirus containing constitutively active FoxO1 (CA-FoxO1) with mutating threonine 24 to alanine, serine 253 to aspartic acid, and serine 356 to alanine. CA-FoxO1 cannot be phosphorylated by insulin or other growth factors. VSMCs were infected with adenovirus containing either GFP or CA-FoxO1, and the expression of *Has2* with or without insulin stimulation was determined by qPCR. The basal level of *Has2* was increased significantly in VSMCs overexpressed with CA-FoxO1. Insulin's effect to induce *Has2* gene expression was inhibited in VSMCs infected with CA-FoxO1 (Fig. 7g). Since CA-FoxO1 can increase Akt phosphorylation to activate mTORC1[32], the increased basal

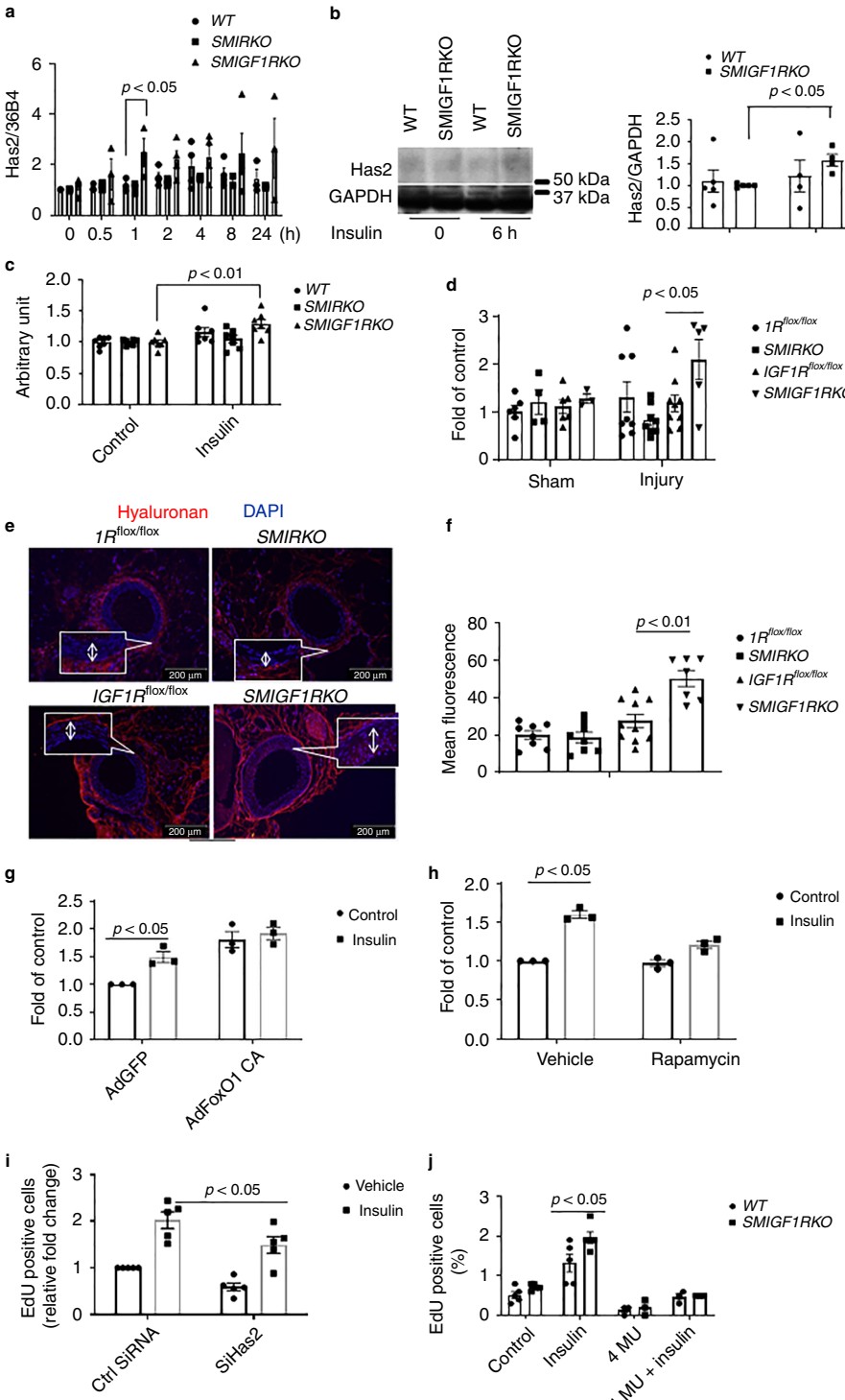

**Fig. 7** *Has2* expression in VSMCs and femoral artery. **a** Starved *WT*, *SMIRKO*, or *SMIGF1RKO* VSMCs were stimulated with 10 nM insulin, and the gene expression of *Has2* was determined by qPCR (0 h, $n = 4$ for each group; 0.5 h, $n = 3$ for each group; 1 h, $n = 3$ for each group; 2 h, $n = 4$ for each group; 4 h and 8 h, $n = 4$ for *WT* and *SMIGF1RKO*, $n = 3$ for *SMIRKO*; 24 h, $n = 3$ for *WT* and *SMIGF1RKO*; $n = 2$ for *SMIRKO*). **b** *Has2* protein expression in VSMCs treated with insulin. **c** Hyaluronan in cultured medium. The hyaluronan content before and 12 h after 10 nM insulin stimulation was determined by ELISA. **d** The femoral arteries were harvested at 9 days after wire injury, and the expression of Has2 gene was determined by qPCR (*IR$^{flox/flox}$* sham, $n = 6$; *IR$^{flox/flox}$* injury $n = 7$; *SMIRKO* sham, $n = 4$; *SMIRKO* injury, $n = 8$; *IGF1R$^{flox/flox}$* sham, $n = 6$; *IGF1R$^{flox/flox}$* injury, $n = 9$; *SMIGF1RKO* sham, $n = 3$; *SMIGF1RKO* injury, $n = 5$). **e, f** Hyaluronan staining in wire-injured femoral artery. Hyaluronan staining was performed on feoral arteries at 4 weeks after wire injury (arrow indicates neointima. *IR$^{flox/flox}$*, $n = 8$; *SMIRKO*, $n = 8$; *IGF1R$^{flox/flox}$*, $n = 9$; *SMIGF1RKO*, $n = 7$). **g** VSMCs were infected with AdFoxO1 CA or AdGFP and then stimulated with insulin for 4 h. The expression of Has2 gene was determined by qPCR ($n = 3$ per group). **h** VSMCs were treated with rapamycin (10 ng/ml) for 1 h and then stimulated with insulin for 4 h. Expression of Has2 gene was determined by qPCR ($n = 3$ per group). **i** The expression of Has2 in VSMCs was knockdown by siRNA, and cellular proliferation was determined by EdU incorporation ($n = 8$ for each group). **j** Starved VSMCs were pretreated with 1 mM 4-MU for 4 h and then stimulated with 100 nM insulin for 24 h. The cell proliferation was determined by measuring EdU-positive cells by flow cytometry ($n = 3$ per group). The data are mean ± SEM. Two-tailed *t* test or two-way ANOVA with a post hoc test. Source data are provided as a Source Data file

level of *Has2* in CA-FoxO1 overexpressed cells could be due to the activation of mTORC1. Thus, we incubated VSMCs with rapamycin, which is a mTORC1 inhibitor. The presence of rapamycin partially inhibited insulin-induced *Has2* gene expression (Fig. 7h).

To demonstrate that *Has2* and hyaluronan play an important role in VSMC proliferation, we reduced *Has2* expression by siRNA and then studied its reduction on cellular proliferation. siRNA decreased *Has2* gene expression by 67% compared with control (Supplementary Fig. 13). Cellular proliferation was decreased significantly by 39% at the basal level and 31% after insulin stimulation in *Has2* knockdown cells (Fig. 7i). The effect of hyaluronan inhibitor 4-MU on VSMC proliferation was studied. As shown in Fig. 7j, insulin (100 nM) greatly elevated Edu incorporation in VSMCs from *SMIGF1RKO* compared with VSMCs from *WT* Mice. The addition of 4-MU significantly inhibited both insulin (Fig. 7j) and 10% fetal bovine serum (Supplementary Fig. 14A) induced Edu incorporation, respectively. To determine the effects of 4-MU on cell death, VSMC were treated with 4-MU for 24 h, and stained cells with Annexin V and PI and assessed by flow cytometry. Treatment with 4-MU increased dead cells (including apoptosis and necrosis) from 18 to 44% (Supplementary Fig. 14B).

## Discussion

Prevailing theory has been that the elevations of circulating IGF-1 and insulin levels in insulin resistance and type 2 diabetes are both partly responsible for the acceleration of VSMC proliferation and restenosis causing stent failure in the treatment for coronary artery disease[15,33]. The major reason for the difficulty in identifying the individual contributions of insulin and IGF1 is the almost complete overlap of cellular signaling and actions of these hormones and their receptors in the VSMC[6,34]. Our findings using specific deletions of IR and IGF1R in VSMC provide the surprising and clear result that the loss of IGF1R does not protect against intimal hyperplasia, whereas deletion of IR decreases VSMC proliferation and intimal hyperplasia following intimal injury. These surprising results strongly suggest that insulin and IR are primarily responsible for the increased risk of restenosis in insulin resistance and diabetes. The findings that insulin and IGF-1 receptors on VSMC proliferation were different is surprising, considering that almost all the biological actions of these hormones are overlapping in most cell types[35]. In fact, most studies have concluded that IGF-1 and IGF1R are the major contributors to VSMC growth due to their greater mitogenic actions at physiological levels than insulin on VSMC cells[6,10]. However, these results suggest that the accelerated restenosis in insulin resistance is mainly due to insulin action via homo-IR receptor and may even be inhibited by IGF1R in VSMCs. The important role of insulin and IR on intimal hyperplasia has been clearly demonstrated in this study by comparing the extent of the intimal hyperplasia to injury in Myh11IRKO vs. SMIGF1KO mice on HFD. This mouse model replicated diabetes and insulin resistance in people by exhibiting elevated IGF-1 and insulin levels, insulin resistance, and enhanced intimal hyperplasia when they are fed HFD vs. control. This pathology of intimal hyperplasia also exhibited predominantly migration and proliferation of VSMC with minor contribution from inflammation, which reflects restenosis clinically[3,4].

In vivo and in vitro studies of VSMC proliferation confirmed the pathological findings that showed the deletion of IR decreased mitogenic effects of insulin. Even more surprising was the finding that insulin's signaling and actions were actually increased to a greater extent in VSMC isolated from *SMIGF1RKO* compared with *WT* mice which contrasted with the decreases in IGF1 effects

in *SMIRKO* cells. In addition, these results suggested that the loss of IGF1 receptor actually enhanced IR cellular effects. These findings indicated that structural differences in IR and IGF1R can mediate different specific actions in VSMC, which is unlike many other cell type that have been studied[36].

The potential mechanisms for the selective and enhanced action of insulin on VSMC with IGF1R deletion was related to the increased binding of insulin to the homo-IR when compared with hetero-IR/IGF1R or homo-IGF1R. In addition, the presence of IGF1R in VSMC may even inhibit insulin and IR actions by the formation of hetero-IR/IGF1 receptors. This conclusion is supported by several studies, including binding experiments, actions in VSMCs containing pure homo-IR or IGF1 receptor in double-knockout cells, and the use of chimera IR/IGF1 receptors. The increased binding of insulin to IR in *SMIGF1RKO* mice or VSMC with *IGF1R* deletion is likely due to the increase of binding affinity, since no changes in IR levels of either α or β subunit of IR were observed at cell surface or at whole-cell level. These results indicated that there are significant differences in the binding properties to homo-IR versus to homo-IGF1 receptors by insulin and IGF-1 to their respective receptors. The additional results from the chimera IR-ECD/IGF1R-ICD versus IGF1R-ECD/IR-ICD receptors in VSMC with double knockout of IR and IGF1R further documented important differences in the binding properties of insulin to IR versus IGF1Rα-binding subunits. The differential enhancement of insulin binding to homo-IR receptors and the inhibition of insulin actions with the addition of both IR-ECD/IGF1R-ICD and IGF1R-ECD/IR-ICD clearly demonstrate the structural differences in the binding region of the IRα subunits versus IGF1Rα subunits especially in the cysteine-rich regions which has been previously noted to be quite different between these two receptors[37,38]. The p-Akt induced by Insulin was slightly higher than that by IGF-1 in IR-ECD/IGF1R-ICD cells, although the difference was not significant. The cells were stimulated with 100 nM insulin or 100 nM IGF1. These data suggest that the extracellular domain of insulin receptor could be fully activated by high dose of IGF-1, whereas the extracellular domain of IGF-1 receptor could only be partially activated by high dose of insulin. Furthermore, the increase of basal activation of Akt in double-chimeric SMCs (Fig. 5e) might be due to autoactivation of double-chimeric receptors by paracrine production of IGF-1. Since double-chimeric receptors formed hybrid chimeric receptor (IR-ECDIGF1R-ECD/IGF1R-ICDIR-ICD), another possibility might be the hybrid chimeric receptor was more sensitive to the trace amount of IGF-1 produced by smooth muscle cells.

The data from the chimera receptor studies also support that the interactions of IRα-binding subunits have synergistic effects in the transmission of insulin signaling from α to β subunit of IR or IGF1R. Our finding in smooth muscle cells are partially different to those observed in preadipocytes using the same chimeric receptor[36], which indicated that the metabolic and growth promoting actions of insulin and IGF-1 receptors are similar but they varied in actions due to differences in interaction between the β subunits of these hormone receptors and intracellular substrates. Our data clearly demonstrate that the α-binding subunit of these receptors can also have major contributions to their cellular actions, especially regarding mitogenic effects of VSMC. The finding of IGF1R could potentially be a negative regulator of insulin action has previously been suggested by Kearney et al., who reported that the presence of IGF1R negatively impacted insulin's regulation of NO actions in the endothelium[39]. Our study clearly demonstrates that the insulin binding is decreased by the formation of hetero-IR/IGF1R, which decreased insulin's signaling in the VSMC resulting in the reduction of VSMC migration and proliferation after injury.

This study was extended further by RNA-seq analysis of insulin-stimulated gene expression in VSMC, which identified *Has2* as a potential downstream target of homo-IR receptors that could be mediating the enhanced insulin actions for VSMC proliferation and intimal hyperplasia. This selective action of IR to increase expression of *Has2* is supported by the findings that insulin increased *Has2* expression in VSMC and in vivo from *SMIGF1RKO* mice to a greater extent than in *WT* VSMC. Thus, we have identified a target of insulin and IR in VSMC. The function of *Has2* is to produce hyaluronan, an important component of the extracellular matrix and critical for VSMC proliferation, migration, and differentiation[40–42]. *Has2* is the major isoform of *Has* in VSMC and has a critical role in cardiovascular tissue formation, since embryotic deletion of *Has2* is lethal with abnormal development of the heart and large vessels[43]. In intimal hyperplasia models, hyaluronan production increases after balloon injury with enhanced levels in the media and even in restenosis of human arteries[44,45]. Our study clearly supports the importance of *Has2* and its actions in VSMC proliferation. In addition, treatment with an inhibitor of Has2 decreases insulin-induced proliferation of VSMC. The mechanism of insulin action to increase *Has2* expression appears to be mediated through multiple pathways, including both AKT and MAPK kinase. Other studies have shown that the direct activation of MAP kinase can regulate Has2 gene expression[46]; however, our findings suggest that p-Akt may also be important.

In summary, we find that insulin through homo-IR receptor is most likely a significant cause for enhancing VSMC proliferation in restenosis. The identification of the selective actions of homo-IR that can be inhibited by IGF1R is exciting especially with the identification of a IR-specific downstream target, Has2, which has already shown to be important for restenosis and possible atherosclerosis. These results suggest inhibition of homo-IR in VSMC at the stent site could be a potential clinical target for treatment in reversing the increase risk of restenosis after stents and angioplasty in people with diabetes and insulin resistance.

## Methods

**Animals**. IR[flox/flox], IGF1R[flox/flox], and IR[flox/flox]/IGF1R[flox/flox] mice were kindly provided by Dr. C. Ronald Kahn (Joslin Diabetes Center, Boston, MA, USA). SM22α Cre and Myh11-Cre mice were obtained from Jackson Laboratory. All of mice have been backcrossed with C56BL/6J for more than six times. Male mice were fed a high-fat diet (Research diet, 60% calories from fat, D12492) starting at 4 weeks of age and continuing for 8 weeks. All protocols for animal use and euthanasia were reviewed and approved by the Animal Care Committee of the Joslin Diabetes Center. They are in accordance with NIH guidelines following the standards established by the Animal Welfare Acts and by the documents entitled "Principles for Use of Animals" and "Guide for the Care and Use of Laboratory Animals".

**Femoral artery wire injury and artery remodeling analysis**. Bilateral wire injury of the femoral artery was performed[14]. In brief, the femoral arteries were injured by three passages of a 0.014-inch-diameter angioplasty guidewire. The femoral arteries were harvested after perfusion with 10% formaldehyde under 75 mmHg pressure. Paraffin-embedded 5- μm sections were stained with the Elastin staining kit (Sigma). Sections were masked for the measurement of the luminal, intimal, and medial areas using NIH ImageJ.

**Smooth muscle cell culture**. Mice aortas were dissected and digested with Dulbecco's modified Eagle medium (DMEM) containing 0.1% collagenase II (Worthington Biochemical Corp, Lakewood, NJ, USA) for 20 min. The adventitia of aorta was stripped off, and the media of aorta was cut into small pieces. Then the aorta pieces were digested with the DMEM containing 0.2% (w/v) collagenase type 1 (Worthington Biochemical Corporation, Lakewood, NJ, USA) and 0.1% BSA at 37 °C for 1 h. After washing and centrifuge, the cells were plated on dishes coated with 0.1% (w/v) collagen I (BioCoat, BD BioSciences, Franklin Lakes, NJ, USA) and were grown in the DMEM with 20% (v/v) fetal bovine serum. The cells were used to do experiments at passages of 2–4.

**Lung endothelial cells culture**. The lung was cut into small pieces and digested with 0.2% collagenase I 1 (Worthington Biochemical Corporation, Lakewood, NJ)

with 0.1% BSA I for 1 h. The digest was pipetted and filtered through a 100 -μm strainer. After centrifuge, the cells were plated into dish coated with 0.2% gelatin. The endothelial cells were purified with anti-ICAM2 antibody (BD Bioscience, San Jose, CA, USA)-conjugated magnet beads twice when the cells were confluent. Endothelial cells were used to do experiments at the passages of 3–4.

**Isolate adipocytes**. Adipocytes were isolated from epididymal fat. The epididymal fat was cut into small pieces and digested with 10% collagenase II in DMEM/0.1% BSA for 20 min. The digest was centrifuged at 300 *g* for 3 min. The adipocytes floating on the top of liquid were collected for western analysis.

**Generate MyH11-CreERT2/IRfox/flox mice**. MyH11-CreERT2 mice were purchased from Jackson Lab (Bar Harbor, ME, USA). IRflox/flox mice were cross-bred with MyH11-CreERT2 mice to generate MyH11-CreERT2/IRfox/flox (MyH11IRKO) mice. Insulin receptor in smooth muscle cells in MyH11IRKO mice was deleted by tamoxifen intraperitoneal injection (Sigma, St. Louis, MO, USA, 50 mg/Kg body weight) once every 24 h for a total of 10 consecutive days. The control mice were IRflox/flox mice which were also received tamoxifen intraperitoneal injection (50 mg/Kg body weight) once every 24 h for a total of 10 consecutive days.

**Western blotting**. Mouse aorta smooth muscle cells were incubated with DMEM containing 0.1% BSA for 48 h and then were stimulated with insulin (Sigma, St. Louis, MO, USA) or IGF1 (Gold Biotechnology Inc., St. Louis, MO, USA) for 15 min. To observe the actions of insulin in vivo, insulin (10 mIU/g body weight) was administered to mice via intravenous injection, and aortas were harvested at 5 min after insulin injection. Tissues were placed in a heavy duty pouch and pulverized on dry ice. The cells or tissues were lysed with RIPA buffer (buffer (50 mM Tris-HCl, 150 mM NaCl, 1% NP-40, 0.5% sodium deoxycholate, and 0.1% SDS). The protein concentration was determined by BCA protein assay reagent (Thermo Scientific, Rockford, IL, USA). Protein samples were separated by electrophoresis in a 7.5% Tris-HCl polyacrylamide gel and transferred to a PVDF membrane, which was blocked with 5% nonfat dry milk in Tris-buffered saline-0.1% Tween-20 and incubated with primary antibody in 4 °C overnight. Detection was carried out using an ECL Plus Western Blotting Detection kit (Thermo Fisher Scientific, Rockford, IL, USA). The membranes were incubated with Restore Plus Western Blot Stripping buffer (Thermo Fisher Scientific, Rockford, IL, USA) for 15 min to remove primary and secondary antibodies. After washing for six times with PBS/0.1% Twen-20, the membranes were blotted with new antibodies. Quantitative densitometry was performed using ImageJ. Antibodies are listed in Table 1.

**Glucose and insulin tolerance tests**. Mice were fasted overnight and then glucose (2 mg/g body weight) or fasted for 6 h, and then insulin (0.75 mIU/g body weight) was administered to mice via intraperitoneal injection. Blood was sampled from tail vein for glucose determination at 0, 30, 60, 120 min after glucose or insulin injection. Glucose was measured by Contour glucometer (Bayer, Mishawaka, IN, USA).

**Blood pressure assessment**. Blood pressure was assessed with the Visitech noninvasive blood pressure analysis system. Up to six mice in each strain group were monitored for 5 min over the course of 5 days. Conscious mice were placed on a heated pad, and a tail cuff was placed in order to get blood pressure measurements.

**Immunofluorescence**. The femoral arteries of mice were subjected to wire injury. After surgery, the mice were intraperitoneally injected with 1 mg Brdu and followed with infusion of Brdu using 7-day osmotic minipump (Alzet, Cupertino, CA, USA) at a rate of 60 μg/day for 7 days. The mice were killed and perfused with 4% PFA after killing and femoral artery was dissected. The femoral artery was embedding in paraffin and 5-μm sections were cut. After blocking and antigen retrieval, the slides were incubated with indicated antibodies. BrdU was stained with a 5-Bromo-2′-deoxy-Uridine Labeling and Detection Kit I (Roche, Indianapolis, IN, USA) following the kit instruction. Anti-SM22α antibody was obtained from Abcam (catalogue number: ab10135) with 1:200 dilution. Second antibody was DyLight 549 Streptavidin (Vector laboratories, Burlingame, CA, USA).

**Elastin staining**. Elastin staining was performed using the Elastic stain kit from Sigma Aldrich (St. Louis, MO, USA). Briefly, deparaffinized slides were stained in Elastic stain solution for 10 min. Then the slides were differentiated in working ferric chloride solution for 3 min. After rinsing in 95% alcohol, the slides were stained with Van Gieson solution for 3 min. Then the slides were rinsed in 95% alcohol and dehydrated to xylene. The images were taken by Olympus FSX100 imaging system and were analyzed by Image J.

**EDU incorporation**. EDU incorporation was measured with an EDU flow-cytometry kit. The cells were starved in the DMEM containg 0.1% BSA for 48 h. Then the cells were stimulated with insulin or IGF-1. After 16 h of stimulation, the cells were incubated with 10 nM EDU (Invitrogen, Carlsbad, CA, USA) for 4 h.

**Table 1 Antibodies information**

| Antibodies | Catalogue number | Source | Concentration |
|---|---|---|---|
| IRβ | 3020s | Cell Signaling Technology | 1:1000 dilution |
| Actin | sc-47778 HRP | Santa Cruz Biotechnology | 1:100,000 dilution |
| IGF1Rβ | 3027 | Cell Signaling Technology | 1:1000 dilution |
| p-Akt | 4060 | Cell Signaling Technology | 1:1000 dilution |
| Akt | 9272 | Cell Signaling Technology | 1:1000 dilution |
| GAPDH | ab9484 | Abcam | 1:2000 dilution |
| p-IGF1R (Tyr 1135/1136)/IR (tyr 1150/1151) | 2969 | Cell Signaling Technology | 1:1000 dilution |
| p-Erk | 9101 | Cell Signaling Technology | 1:1000 dilution |
| Erk | 4695 | Cell Signaling Technology | 1:1000 dilution |
| Pan-cadherin | sc-10733 | Santa Cruz Biotechnology | 1:1000 dilution |
| Has2 (C-5) | sc-365263 | Santa Cruz Biotechnology | 1:500 dilution |
| Anti-mouse IgG | sc-516102 | Santa Cruz Biotechnology | 1:5000 dilution |
| Anti-rabbit IgG | sc-2357 | Santa Cruz Biotechnology | 1:5000 dilution |

Then the cells were harvested and fixed with fixation buffer. After washing and permeabilization buffer, the cells were incubated with Click-iT® EdU reaction cocktail for 30 min. After washing, the EDU-positive cells were measured by flow cytometer (BD LSR II). For 4-MU treatment, after starving with DMEM containg 0.1% BSA for 48 h, the cells were incubated with 1 mM 4-MU for 4 h. Then the medium was replaced with new DMEM containing 1 mM 4-MU.

**RNA sequencing**. VSMCs were cultured from aortas of IGF1Rflox/flox mice or SMIGF1RKO mice. VSMCs from SMIGF1RKO mice were infected with AdCre to further knockdown IGF1R. VSMCs from IGF1Rflox/flox mice were infected with AdGFP. The VSMCs were starved with 0.1% BSA for 48 h and then stimulated with insulin for 4 h. RNA was extracted with the Purelink RNA mini kit (The life technologies, Carsbad, CA, USA) and 100 ng of RNA was used to create library for RNA sequencing ($n = 3$ for each group). RNA sequencing was performed using Illumina NextSeq 500- Single Read 75 (SR75) by the Center for Cancer Computational Biology (CCCB) at Dana Farber Cancer Institute of Harvard Medical School. Sequencing reads from FASTQ files were aligned to the reference genome using the RNA-specific STAR aligner to generate sequence alignment (BAM) files. The featureCounts tool was then used taking in the BAM files to count sequencing reads mapping to the reference genome at the gene level. The read counts were subsequently normalized between samples using DESeq. The normalized read-count file was further used in RNA-Seq analysis. RNA sequencing data were analyzed with limma, an R package for linear modeling that powers differential expression analyses. We controlled batch effects and used sample quality weights.

**Real-time PCR**. One microgram of mRNA was used to generate cDNA using the high-capacity cDNA reverse transcription kit (Applied Biosystems, Grand Island, NY, USA). Gene expression level was normalized to the expression level of 36B4.
   The primer sequences are listed in Table 2.

**Insulin and IGF-1 binding**. Cells were incubated with $I^{125}$-labeled insulin or IGF-1 (PerkinElmer, Waltham, MA, USA) with indicated concentration of unlabeled insulin or IGF-1 on ice for 5 h. After washing with cold PBS for three times, the cells were lysed with 4% NaOH for 1 h at room temperature. Then the cell lysis was transferred to the scintillation bottle, and the radioactivity was measured by a scintillation counter.

**Generation of adenovirus containing IR-ECD/IGF1R-ICD or IGF1R-ECD/IR-ICD chimeric receptor**. Plasmids containing IR-ECD/IGF1R-ICD or IGF1R-ECD/IR-ICD chimeric receptor were kindly provided by Dr. C. Ronald Kahn (Joslin Diabetes Center, Boston, MA, USA). The chimeric receptors were cloned into adenovirus backbone plasmid using ViraPower Adenoviral Gateway Expression kit (Invitrogen, Carlsbad, CA, USA). The VSMCs were infected with adenovirus at 100 MOI.

**Measurement of hyaluronan in medium**. The concentration of hyaluronan in cell cultured medium was determined by the hyaluronan ELISA kit (R&D systems, Minneapolis, MN, USA).

**Hyaluronan staining**. The sections were incubated with biotin-conjugated hyaluronic acid-binding protein (8 μg/ml, Fisher, Waltham, MA, USA) at 4 °C overnight. After washing, the slides were incubated with avidin-conjugated dylight 549.

**SiRNA transfection**. Control siRNA (The Silencer Select Negative Control No. 1 siRNA) or HAS2 siRNA (#s67368, sense, 5′-GACCUACCCUGGGAUUAAAtt-3′; antisense, 5′-UUUAAUCCCAGGGUAGGUCag-3′) were predesigned by Ambion (Foster City, CA, USA). 10 nM siRNAs were transfected to VSMCs using

**Table 2 Primer information**

| 36b4 UP | CCGGATGTGAGGCAGCAG |
|---|---|
| 36b4 DN | GCTCCAAGCAGATGCAGCA |
| IR UP | GGAAGCTACATCTGATTCGAGG |
| IR DN | TGAGTGATGGTGAGGTTGTGT |
| Egln3 UP | GAAGGGAAATCGTTTGTAGCAG |
| Egln3 DN | GAAGTACCAGACAGTCATAGCG |
| Sipr3 UP | AGAGTGTCATTTCCCAAGACTG |
| Sipr3 DN | ACTCTCCCAATTGTTCCCTG |
| Slc16a3 UP | CCTGTCATGCTTGTGGGTG |
| Slc16a3 DN | GGAAGGCTGGAAGTTGAGAG |
| Rcor2 UP | GCATGTACCTGAGTCCTGAAG |
| Rcor2 DN | CTGCTATTGGTCTGCTTCATG |
| Adamts14 UP | AACTCTTCCGACAACCCTTG |
| Adamts14 DN | GTAGTCTGAGTTGTCTGTGCG |
| Pfkl UP | GGAAAGCCTATCTCATCCAGC |
| Pfkl DN | CCATACCCATCTTGCTACTCA |
| Pfkfb3 UP | CTGACTCGCTACCTCAACTG |
| Pfkfb3 DN | ACTGTTTTCGGACTCTCATGG |
| Relt UP | CCCTGTCTTGCCTCTTTGTG |
| Relt DN | CATAATGTGCCCTGTCCTGG |
| Gys1 UP | CCATGTCTTCACTACCGTATCC |
| Gys1 DN | CTCTGAGCATGAAGGTTCTGG |
| Has2 UP | AGTCATGTACACAGCCTTCAG |
| Has2 DN | CTCCAACACCTCCAACCATAG |
| Kif26b UP | TCCCACAAGACGCTTCTCAG |
| Kif26b DN | TCGTCCCTTCCGATCATGGT |
| Trabd2b UP | GATGAAGATGTGCTGAGAGGG |
| Trabd2b DN | CTAGGGATGTGGAAGTTGTCTC |
| IGF1R UP | GTACCGCATCGATATCCACAG |
| IGF1R DN | ATGGAGTTTTCGGGTCTT |

Lipofectamine RNAiMAX from Invitrogen (Carlsbad, CA, USA) according to the manufacturer's instructions. Knockdown efficiency was confirmed after 24 h and 72 h transfection with siRNAs.

**Statistics**. Measurements are from distinct samples. Comparisons of the two groups were made using paired or unpaired $t$ test as appropriate. Comparison among more than two groups was performed by one-way ANOVA or two-way ANOVA followed by the post hoc analysis to evaluate statistical significance between the two groups. Statistical significance was defined as $p < 0.05$. In text and graphs, the data are presented as the mean ± standard error.

**Reporting summary**. Further information on research design is available in the Nature Research Reporting Summary linked to this article.

## Data availability
The data sets generated during and/or analyzed during this study are available from the corresponding author on reasonable request. The source data underlying Figs. 1–7 and Supplementary Figs. 1–14 are provided as a Source Data file. The RNA sequencing data

set (normalized counts) is provided as a Supplementary Dataset in the excel format. The BAM or FASTQ files were not saved before the closing of the core at DFCI, hence the limitations to the accessibility of the raw data.

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

## Acknowledgements

This study was supported by NIH grant R01-DK-053105. We are grateful for the expert technical assistance provided by Biostatistic and Bioinformatic Core, Advanced Microscopy Core, Flow Cytometry Core, Mouse Physiology Core, Genomic Core, which are supported by NIH grants 5P30DK036836 and S10OD021740. We thank Dr. Kevin Croce for the technique support in femoral artery wire injury. The content of this paper is solely the responsibility of the authors, and does not necessarily represent the official views of the funding agencies.

## Author contributions

Q.L. performed experiments, analyzed the data, and wrote the paper. J.F., Y.X., W.Q., Q.H. and W.C. performed experiments. A.I, K.P., J.F., H.Y., C.R.K. and C.R.M. participated in the discussions and critical review of the paper; G.L.K. supervised the studies and wrote the paper.

## Competing interests

The authors declare no competing interests.
