## [Peer Review File · Nature Communications]

Reviewers' comments:

Reviewer #1 (Remarks to the Author):

Using multiple sophisticated approaches, this piece offers substantive new knowledge that builds constructively on existing literature, though some elements of the manuscript would benefit from clarification and more methodologic detail.

Major

1. Methods (fine to place in Supplement) need more detail for readership and subsequent replication:
 - a. Source of primary antibodies should be listed, as well as any retrieval methods
 - b. More details on tissue harvest in the femoral wire injury model—was there perfusion fixation? This is important since for instance in Figure 2B there is an intimal differential between groups that is lost when normalized (via I/M ratio) to medial area (Figure 2C; lines 146-149 of the text should be corrected since statistically the groups are the same).
 - c. Protein isolation/purification procedures (including from the tissues reported in Supplemental Figure 1C, etc.) should be delineated.
2. Did the VSMC numbers differ in the murine intimal lesions? If increased in SMIGF1RKO mice, did this correlate with a decrease in collagen content?
3. Microscopic images in Figures 1A and 2A are unacceptable. All need repair of white balance, and inclusion of reticules and additional images showing a higher magnification.
4. Please provide higher magnification images for Figure 7E. Is the positive staining truly in the intima or rather in the adventitia? This reviewer was unable to tell from the evidence provided.

Minor

1. Lines 63, 254: Is AIH “arterial intimal hyperplasia”?

2. Gel labels truncated in Supplemental Figure 1B.
3. “Stainng” typo in the legend of Figure 2.
4. Since animal numbers in treatment groups were small, dot plots may be more informative to convey variation within and between groups.

Reviewer #2 (Remarks to the Author):

The manuscript titled “Receptors for insulin, not IGF-1, accelerate intimal hyperplasia by upregulation of hyaluronan synthase 2” by Qian Li, et. al. demonstrates opposing actions of insulin-IR signaling and IGF1-IGF1R signaling on injury-induced neointima formation and SMC proliferation. They demonstrate that SMC-specific KO of IR reduces, but SMC-specific KO of IGF1R exacerbates neointima formation. Using molecular approaches and clever receptor chimera approaches, the authors demonstrate that enhanced effects mediated by IGF1R KO are through increased IR homodimer formation and insulin actions. This is potentially a very interesting manuscript as it uncovers distinct and opposing actions of these signaling partners, previously believed to exert similar effects. There are some major and more minor points that should be addressed, which would significantly strengthen the manuscript, as outlined below. Importantly, throughout the manuscript the authors state changes, but there is no statistical significance often times, or there is statistical significance, but the overall changes are very modest suggesting that the biological significance is questionable. The two mouse models are very different, using different Cre drivers, which is a major concern. Finally, the signaling and molecular mechanism underlying the overall conclusions were not forcefully addressed.

Much more detail is required in the Methods and Supplemental Materials and Methods section. All information should be included in the Supplement and reiterated in abbreviated form in the body of the paper: description of mice and mouse procedures; cultured SMC passage number; whole cell lysate isolation (buffer); Western primary antibody clone numbers and concentration used and secondary antibody; glucose measurements conducted using what method; immunohistochemistry must be expanded (antibodies used, clone numbers, concentrations, secondary antibodies or detection system); immunofluorescent staining was not addressed; microscopes used for image analysis should be included and software used for quantification; elastin staining must be elaborated; EdU flow cytometry using what flow cytometer, software; qPCR must be expanded to include RNA isolation method, gene(s) analyzed, primer sequences provided; adenovirus MOI should

be included; there is absolutely no information regarding the RNA-Seq analysis (e.g. how many samples/repetitions, statistical analysis).

Line 63: please define AIH.

Supplemental figure 1: how were SMCs, adipocytes, and ECs isolated for Western analysis?

Supplemental Figure 2: please show actin loading levels in Western blots.

Line 119: Was BrdU or EdU used to measure proliferation?

Lines 123-125, Figure 1H: please report EdU changes as fold increases. How were the data analyzed for this figure? ANOVA? If so, are insulin/IGF1-stimulated SMIRKO SMCs different than control?

Why were SM22-Cre mice used to generate IR KO mice, but Myh11-Cre mice used to generate IGF1R KO mice? Since these transgenic mice are constitutive and SM22 is transiently expressed in cardiomyocytes, but Myh11 is specific to SMCs, cardiomyocyte-specific IR KO could be mediating some of the observed effects through potential paracrine mechanisms. This would make interpretation of the data somewhat misleading and the two models are not synonymous.

Immunofluorescence images showing SM22 and BrdU double staining are of relative poor quality. Please show higher power images (insets from low power).

Lines 146-146, Figure 2: The text states that I/M ratio and medial area are increased in IGF1R KO mice, but statistics show that this is not the case. Please state that I/M ratio and medial area were not different or at the minimum, there was a trend toward an increase, but this was not statistically significant.

Molecular weight markers need to be added to all Western blots.

Line 171: "SMIGF1RKO VSMCs, insulin at 1nM stimulated pTyr-IR, which was not observed in WT VSMCs." This was not statistically significant.

Figure 3: Why such a dramatic increase in IGF1R phosphorylation in IGF1R KO SMCs in response to IGF1 treatment?

Line 184: please report pAkt as fold change, not percent change. In general, the authors use both percent change and fold change throughout the manuscript. It would strengthen the manuscript if this is kept consistent.

Figure 7: while Has2 appears to be upregulated in IGF1R KO SMCs, the data shown are underwhelming with only 1.5-fold or less changes. Therefore there are concerns regarding the biological significance of these findings. The in vivo immunofluorescence images are of poor quality and difficult to appreciate the changes. Are the changes observed selectively in SMCs? Panels F&H are not addressed in the legend. What is the significance of 4-MU-mediated reduction of FBS-induced proliferation? Is it possible that 4-MU is inducing cell death? A molecular approach, whereby Has2 is depleted in SMCs would strengthen this finding.

Line 296: panel E shows IRflox/flox, SMIRKO, IGF1Rflox/flox, and SMIGF1RKO mice. What are the WT mice that ar3e being referred to (“compared to WT as well as to IR flox/flox and SMIRKO mice”)?

Reviewer #3 (Remarks to the Author):

The authors in this manuscript addressed the role of insulin receptor (IR) and IGF-1 in control of smooth muscle and vascular biology in mouse models, in particular on arterial intima hyperplasia and restenosis after angioplasty, especially in diabetes. Using the insulin receptor (SMIRKO) or IGF-1 receptor (SMIGF1RKO) knockout in VSMC with high fat diet, the authors showed that intima hyperplasia was attenuated in SMIRKO mice, but exacerbated in SMIGF1RKO mice. In VSMC, deleting IGF1R increased homodimers of IR, enhanced insulin binding, stimulated p-Akt and proliferation, but deleting IR decreased responses to insulin and IGF-1. Studies using chimera of IR(extracellular

domain)/IGF1R(intracellular-domain) or IGF1R(extracellular domain)/IR(intracellular-domain) demonstrated homo-dimer IR α enhanced insulin binding and signaling which was inhibited by IGF1R α . RNA-seq analysis further identified hyaluronan synthase2, as a novel target of homo-IR, with its expression increased by IR activation and intima hyperplasia in SMIGF1RKO mice and decreased in SMIRKO mice. Enhanced intima hyperplasia in diabetes is mainly due to insulin signaling via homo-IR, associated with increased Has2 expression.

This study is very important and the conclusion is solid to address the role of insulin receptor and IGF-1 in the smooth muscle cells and vascular organ in vivo. The manuscript is well written and experiments were well designed and conclusion is novel. I only have some concerns that the author can help interpret and enhance the quality manuscript.

1) As we know that insulin activates both Akt and Erk pathway via the insulin receptor or IGF-1R. This is supported by the figure 3A where insulin-induced both Akt and ERK phosphorylation were reduced in IRKO cells. What is other possible mechanism for insulin-induced Akt and ERK phosphorylation in the IRKO cells? Is that possible that IGF1R can play a role in mediating insulin action when IR is absent?

2) If IGF1R or IGF1 serves as a competing inhibitor for IR signaling in activation of Akt as authors suggested, whether IGF1R antagonist or blocker of blood IGF1 can be developed as an insulin sensitizer for metabolic control, such as blood glucose? Also, when we pursue insulin sensitizer for glucose and metabolic control by increasing IR signaling, how much the enhancing IR signaling can have potential detrimental effect on arterial intima hyperplasia, as seen in IGF1RKO mice? This should be further discussed.

Reviewer 1

Major

1. Methods (fine to place in Supplement) need more detail for readership and subsequent replication:

a. Source of primary antibodies should be listed, as well as any retrieval methods

We thanked the reviewer for the advice. The sources of various antibodies have been added in Supplemental Methods (Page 3). The retrieval method is described in Supplemental Methods (Page 2).

b. More details on tissue harvest in the femoral wire injury model—was there perfusion fixation? This is important since for instance in Figure 2B there is an intimal differential between groups that is lost when normalized (via I/M ratio) to medial area (Figure 2C; lines 146-149 of the text should be corrected since statistically the groups are the same).

We thanked the reviewer for the comments. Perfusion fixation was performed with 75 mmHg pressure before harvesting tissues. The description in lines 146-149 has been corrected and the new description is “I/M ratio was not different between WT and SMIGF1RKO mice”.

c. Protein isolation/purification procedures (including from the tissues reported in Supplemental Figure 1C, etc.) should be delineated.

Cultured cells were washed twice with ice-cold PBS and lysed in RIPA buffer (50 mM Tris-HCl, 150 mM NaCl, 1% NP-40, 0.5% sodium deoxycholate, and 0.1% SDS) containing proteinase inhibitor. Tissues were placed in a heavy duty pouch and pulverized on dry ice. Then RIPA buffer containing proteinase inhibitor was added into the pouch to lyse the tissues. Protein concentration was determined by BCA protein assay reagent. This description is now in Supplemental Methods Page 2.

2. Did the VSMC numbers differ in the murine intimal lesions? If increased in SMIGF1RKO mice, did this correlate with a decrease in collagen content?

VSMC numbers and the collagen content in the femoral artery intimal lesions were determined. Sections of wire-injured femoral artery from WT or SMIGF1RKO mice were stained with VSMC marker SM22 α or macrophages marker Mac2. VSMC numbers in the intimal area of femoral artery of SMIGF1RKO mice were increased by 50% compared to that of WT mice. We also performed collagen staining using Trichrome staining with the collagen stained blue. The collagen content in the intimal area was not different between two groups of mice. However, hyaluronan was increased in the intima of SMIGF1RKO mice compared to WT Mice. Thus, the increased intimal hyperplasia in SMIGF1RKO mice was due to increased VSMCs number and hyaluronan content. This is consistent with increased expression of Has2, which produces hyaluronan and promote cell proliferation. These new data are described in lines 151-155 on page 7 in Result Section and shown in Supplemental Figure 6.

3. Microscopic images in Figures 1A and 2A are unacceptable. All need repair of white balance, and inclusion of reticules and additional images showing a higher magnification.

We thanked the review for the advice. The white balance has been corrected on images in Figures 1A and 2A and higher magnification images are provided.

4. Please provide higher magnification images for Figure 7E. Is the positive staining truly in the intima or rather in the adventitia? This reviewer was unable to tell from the evidence provided.

We thanked the reviewer for this comments. Higher magnification images for Figure 7E are provided. We quantitated the staining of hyaluronan in intimal area, which is shown in Figure7F.

Minor

1. Lines 63, 254: Is AIH “arterial intimal hyperplasia”?

Yes, AIH is arterial intimal hyperplasia. We defined it in Line 63.

2. Gel labels truncated in Supplemental Figure 1B.

Truncated gel in Supplemental Figure1B has been corrected.

3. “Staining” typo in the legend of Figure 2.

This typo has been corrected.

4. Since animal numbers in treatment groups were small, dot plots may be more informative to convey variation within and between groups.

Thank you for this comment. Figure1B-D and Figure 2B-D are now shown as dot plots.

Reviewer #2 (Remarks to the Author):

Major

1. Importantly, throughout the manuscript the authors state changes, but there is no statistical significance often times, or there is statistical significance, but the overall changes are very modest suggesting that the biological significance is questionable.

We thanked the reviewer for these comments. Since Has2 is an enzyme, a modest change of its protein expression could potentially cause a significant change of downstream outcomes. To test the effects of how a modest change in Has2 expression can affect cellular proliferation, we knocked down Has2 expression by 67% using siRNA in VSMC. This resulted in a decrease of cells proliferation by 39% at basal level and 31% after insulin stimulation in Has2 knockdown cells. The new data is shown in Figure 7I and Supplemental Figure 12 and described in lines 342-347 on page 20 of the Result Section. Please also see our reply to the Editor on this point.

2. The two mouse models are very different, using different Cre drivers, which is a major concern.

We thanked the reviewer for raising this very important question. We have generated a new mice line in which insulin receptor in VSMCs was deleted by Myh11 driving Cre (Myh11IRKO). The mice were subjected to femoral artery injury and intimal hyperplasia was determined by elastin staining. The intimal area and intima/media ratio were significantly decreased by 49% and 60%, respectively, in the femoral artery of Myh11IRKO mice compared to WT Mice after wire injury. The thickness of arterial media was not different between two groups of mice. The new data is shown in Supplemental Figure 7 and described in lines 162-170 on page 7 of the Result Section.

3. Finally, the signaling and molecular mechanism underlying the overall conclusions were not forcefully addressed.

Thank you for the comments regarding the need for more data of the signaling and molecular mechanisms on the regulation of Has2 expression. Our results showed that insulin regulates Has2 expression in VSMC through activation of Akt. The new data (Figure 7G-H) explored the roles of FoxO and mTOR, the two major down-stream targets of Akt. Insulin increased phosphorylation of FoxO1 at threonine 24 and FoxO3a at threonine 32 (Supplemental Figure 11). Recently it was reported that FoxO1 could bind to the promoter of Has2 and inhibit the transcription of Has2¹. Thus, we assessed the effect of insulin to regulate Has2 gene expression by infecting VSMCs with adenovirus containing constitutively active FoxO1 (CA-FoxO1) with mutating threonine 24 to alanine, serine 253 to aspartic acid and serine 356 to alanine. CA-FoxO1 cannot be phosphorylated by insulin or other growth factors. VSMCs were infected with adenovirus containing either GFP or CA-FoxO1 and the expression of Has2 with or without insulin stimulation was determined by qPCR. The basal level of Has2 was increased significantly in VSMCs overexpressed with CA-FoxO1. Insulin's effect to induce Has2 gene expression was inhibited in VSMCs infected with CA-FoxO1 (Figure 7G). Since CA-FoxO1 can increase Akt phosphorylation to activate mTORC1², the increased basal level of Has2 in CA-FoxO1 overexpressed cells could be due to the activation of mTORC1. Thus, we further treated VSMCs with rapamycin, which is a mTORC1 inhibitor. The presence of rapamycin partially inhibited insulin induced Has2 gene expression (Figure 7H). These findings suggest that insulin regulates Has2 gene expression through both FoxO and mTORC1 pathway. The new data are shown in Figure 7 G-H and Supplemental Figure 11 and described in lines 325-340 on page 19 of the Result Section.

Minor:

1. Much more detail is required in the Methods and Supplemental Materials and Methods section. All information should be included in the Supplement and reiterated in abbreviated form in the body of the paper: description of mice and mouse procedures; cultured SMC passage number; whole cell lysate isolation (buffer); Western primary antibody clone numbers and concentration used and secondary antibody; glucose measurements conducted using what method; immunohistochemistry must be expanded (antibodies used, clone numbers, concentrations, secondary antibodies or detection system); immunofluorescent staining was not addressed; microscopes used for image analysis should be included and software used for quantification; elastin staining must be elaborated; EdU flow cytometry using what flow cytometer, software; qPCR must be expanded to include RNA isolation method, gene(s) analyzed, primer sequences provided; adenovirus MOI should be included; there is absolutely no information regarding the RNA-Seq analysis (e.g. how many samples/repetitions, statistical analysis).

We thanked the reviewer's advice. All of the reviewer's suggestions had been addressed and detailed Methods are described in Supplemental Documents. Detailed explanation of the methods are described in answers to Reviewer 1.

2. Line 63: please define AIH.

AIH is artery intimal hyperplasia. We defined it in line 63.

3. Supplemental figure 1: how were SMCs, adipocytes, and ECs isolated for Western analysis?

Detailed methods for isolating SMCs, adipocytes and ECs are described in Supplemental Methods. Briefly, SMCs were cultured from thoracic aorta. The thoracic aorta was dissected and digested with 0.1% collagenase II in DMEM/0.1%BSA at 37°C for 20 minutes. Then adventitia of the aorta was stripped off and the aortic media was cut into small pieces and further digested with 0.2% collagenase I in DMEM/0.1%BSA at 37°C for 1 hour. The digested aortic media was plated on 0.2% gelatin coated plate. The VSMCs at passage 2-4 were used for all experiments. Adipocytes were isolated from epididymal fat, which were cut into small pieces and digested in 10% collagenase II in DMEM/0.1%BSA at 37°C for 20 minutes. The digest was centrifuge at 300 g for 3 minutes. The adipocytes floating on the top of tube were collected for Western analysis. Endothelial cells were cultured from mouse lung which was cut into small pieces and digested with 0.2% collagenase I/0.1%BSA at 37°C for 1 hour. The digest was pipetted and filtered through 100 µm strainer. After centrifugation, cells were plated into dish coated with 0.2% gelatin. Endothelial cells were purified with anti-ICAM2 antibody conjugated magnet beads twice. The purity of endothelial cells was determined by morphology and ICAM2 expression. Endothelial cells were used for experiments at the passage of 3-4. These descriptions have been placed in Supplemental Method Section on page 1.

4. Supplemental Figure 2: please show actin loading levels in Western blots.

The levels of actin are now shown in Supplemental Figure 2.

5. Line 119: Was BrdU or EdU used to measure proliferation?

We used BrdU to measure proliferation in wire-injured femoral arteries in vivo and EdU to measure proliferation in cultured cells. The data described in Line 119 was proliferation in wire-injured femoral artery and BrdU was used for measuring proliferation. The femoral arteries of mice were subjected to wire injury and then BrdU was infused into mice with osmotic minipump. The femoral arteries were harvested at one week after wire injury. Sections of femoral artery were stained with antibodies

recognizing BrdU and SM22 α . Then VSMCs proliferation was determined by counting BrdU and SM22 α double positive cells. All of these details are described in various Figure legends.

6. Lines 123-125, Figure 1H: please report EdU changes as fold increases. How were the data analyzed for this figure? ANOVA? If so, are insulin/IGF1-stimulated SMIRKO SMCs different than control?

We changed Figure 1H and reported the data as fold increases. The data was analyzed by two-way ANOVA. Insulin induced proliferation was not different compared to control in SMIRKO VSMCs. IGF1 significantly increased VSMCs proliferation in SMIRKO compared to WT mice. These results are described in line 127 on page 4 of Result Section.

7. Why were SM22-Cre mice used to generate IR KO mice, but Myh11-Cre mice used to generate IGF1R KO mice? Since these transgenic mice are constitutive and SM22 is transiently expressed in cardiomyocytes, but Myh11 is specific to SMCs, cardiomyocyte-specific IR KO could be mediating some of the observed effects through potential paracrine mechanisms. This would make interpretation of the data somewhat misleading and the two models are not synonymous.

SM22-Cre was more efficient than Myh11-Cre in deletion of floxed genes. IR was decreased by more than 95% in VSMCs when Cre was driven by SM22 promoter. However, SM22 Cre was also expressed in heart. Thus we cross-bred IGF1R $flox/flox$ mice with Myh11-Cre mice. IGF1R expression was decreased by 70% in VSMCs of Myh11-Cre/IGF1R $flox/flox$ mice without affecting IGF1R in the heart. Now we have generated Myh11 Cre driving VSMC specific IR knockout (Myh11IRKO) mice and performed wire injury on the femoral artery of these mice. The intimal area and intima/media ratio were decreased significantly by 49% and 60%, respectively, in the femoral artery of Myh11CreIRKO mice after wire injury compared to WT mice. The new data are shown in Supplemental Figure 7A+B and described in lines 162-170 on page 7 of Result Section.

8. Immunofluorescence images showing SM22 and BrdU double staining are of relative poor quality. Please show higher power images (insets from low power).

Higher magnification images are now provided as shown in Figure 1E and 2E.

9. Lines 146-146, Figure 2: The text states that I/M ratio and medial area are increased in IGF1R KO mice, but statistics show that this is not the case. Please state that I/M ratio and medial area were not different or at the minimum, there was a trend toward an increase, but this was not statistically significant.

We have revised the text to “I/M ratio and medial area were not significantly different between WT and IGF1RKO mice”.

10. Molecular weight markers need to be added to all Western blots.

Molecular weight markers have been added to all Western blots.

11. Line 171: “SMIGF1RKO VSMCs, insulin at 1nM stimulated pTyr-IR, which was not observed in WT VSMCs.” This was not statistically significant.

We have restated as follows: “Insulin stimulated pTyr-IR in a dose dependent manner from 1 nM to 100 nM in both WT and SMIGF1RKO mice, but the effects of insulin were more pronounced in SMIGF1RKO mice than WT mice.” which is now in lines 184-186 on page 9 of Result Section.

12. Figure 3: Why such a dramatic increase in IGF1R phosphorylation in IGF1R KO SMCs in response to IGF1 treatment?

We thanked the reviewer for this very good question. IGF1 stimulated IGF1R phosphorylation at 10-100 nM in WT VSMCs, but its effect was much weaker in SMIGF1RKO VSMCs. IGF1R phosphorylation was decreased by 50% in SMIGF1RKO VSMCs compared to WT VSMCs when cells were stimulated with 100 nM IGF1. The possible explanation for IGF1's effect to induce IGF1R tyrosine phosphorylation even when the residual IGF1R is <30% is that IGF1 at 100nM is at pharmaceutical level and can activate all the remaining IGF1R to the maximum level. The phosphorylation of IGF1R by 100 nM IGF1 in SMIGF1RKO VSMCs might be due to the phosphorylation of residual IGF1R.

13. Line 184: please report pAkt as fold change, not percent change. In general, the authors use both percent change and fold change throughout the manuscript. It would strengthen the manuscript if this is kept consistent.

The changes for pAkt are now reported as fold increases throughout the manuscript.

14. Figure 7: while Has2 appears to be up-regulated in IGF1R KO SMCs, the data shown are underwhelming with only 1.5-fold or less changes. Therefore there are concerns regarding the biological significance of these findings.

We thanked the reviewer on the biological significance of a change of Has 2 expression of 50%. Analysis of the injured femoral artery showed a parallel increase of Has2 by 50% and hyaluroan of 2 fold in the intima of femoral artery of SMIGF1RKO mice after wire injury (Figure 7E-F). We have also knocked down Has2 expression by 67% in VSMCs with a decrease of cellular proliferation (EdU) by 39% at basal level and 31% after insulin stimulation compared to WT Cells. These new data suggest that a decrease of 50% of Has2 is biologically significant. The new data are shown in Figure 7I and Supplemental Figure 12 and described in Result Section in lines 342-347 on page 20.

15. The in vivo immunofluorescence images are of poor quality and difficult to appreciate the changes. Are the changes observed selectively in SMCs?

We thanked the reviewer for pointing out this. We have revised the images with higher magnification images of Figure 7E. We have quantitated hyaluronan in the intima of artery. No clearly discernible changes of hyaluronan were noted in the media or the adventitia. See lines 315-317 on page 18 of Result Section.

16. Panels F&H are not addressed in the legend.

The description for Figure F and H are now added in the Figure legend.

17. What is the significance of 4-MU-mediated reduction of FBS-induced proliferation? Is it possible that 4-MU is inducing cell death? A molecular approach, whereby Has2 is depleted in SMCs would strengthen this finding.

Multiple reports have shown that hyaluronan is important for cell growth. Blocking the production of hyaluronan not only can affect insulin induced cell proliferation, but it will also inhibit cell proliferation

by other growth factors. This conclusion suggested that 4-MU also inhibited the growth effect of FBS. We found blocking the production of hyaluronan could dramatically inhibit FBS induced cell proliferation.

To answer the Reviewers questions on whether 4-MU induced cell death, we treated VSMC with 4-MU for 24 hours and stained cells with Annexin V and PI, as markers of cell death. Treatment with 4-MU increased percent of dead cells (including apoptosis and necrosis) from 18% to 44%. These new data are shown in Supplemental Figure 13 and described in lines 351-354 on page 20 of Result Section.

As described above, we knocked down Has2 by siRNA which decreased Has2 expression by 67%. Then insulin-induced cell proliferation was determined by EdU incorporation and quantitated by flow cytometry. Cell proliferation induced by insulin as measured by EdU incorporation was also significantly decreased by 39% at basal level and 31% after insulin stimulation in Has2 knock down cells. The new data is shown in Figure 7I and described in lines 342-347 on page 20 of Result Section.

18. Line 296: panel E shows IRflox/flox, SMIRKO, IGF1Rflox/flox, and SMIGF1RKO mice. What are the WT mice that are being referred to (“compared to WT as well as to IR flox/flox and SMIRKO mice”)?

WT Mice used were IGF1R flox/flox. We have revised the sentence to “compared to WT (IGF1R flox/flox) as well as to IR flox/flox and SMIRKO mice.”

Reviewer #3 (Remarks to the Author):

This study is very important and the conclusion is solid to address the role of insulin receptor and IGF-1 in the smooth muscle cells and vascular organ in vivo. The manuscript is well written and experiments were well designed and conclusion is novel. I only have some concerns that the author can help interpret and enhance the quality manuscript.

1) As we know that insulin activates both Akt and Erk pathway via the insulin receptor or IGF-1R. This is supported by the figure 3A where insulin-induced both Akt and ERK phosphorylation were reduced in IRKO cells. What is other possible mechanism for insulin-induced Akt and ERK phosphorylation in the IRKO cells? Is that possible that IGF1R can play a role in mediating insulin action when IR is absent?

We thank Reviewer 3 for this important question. Insulin could still induce Akt and Erk phosphorylation at 100 nM in SMIRKO VSMCs although insulin receptors were completely deleted. However, it is likely that the high concentration of insulin (100 nM) can activate both IR and IGF1R. Thus, we believe, insulin at 100 nM is inducing pAkt and pErk mostly via IGF1R. Please see description in lines 454-457 on Page 5 of Discussion Section.

2) If IGF1R or IGF1 serves as a competing inhibitor for IR signaling in activation of Akt as authors suggested, whether IGF1R antagonist or blocker of blood IGF1 can be developed as an insulin sensitizer for metabolic control, such as blood glucose? Also, when we pursue insulin sensitizer for glucose and metabolic control by increasing IR signaling, how much the enhancing IR signaling can have potential detrimental effect on arterial intima hyperplasia, as seen in IGF1RKO mice? This should be further discussed.

It has been reported that IGF1R inhibited insulin signaling by forming hybrid IR/IGF1R in skeletal muscle. So it is possible that reducing the expression of IGF1R in skeletal muscle will enhance insulin action in skeletal muscle and improve glucose control.

Our findings suggest that enhancing insulin actions in VSMCs may enhance protein synthesis and proliferation. However, this effect may only be important for intimal hyperplasia of artery such as in arterial stent procedures. Thus, our findings suggest that inhibitors to Has 2 could be added to drug-eluting stents in order to reduce restenosis and stent failure in people with diabetes.

Reference

- 1 Liu, S. & Cheng, C. Akt Signaling Is Sustained by a CD44 Splice Isoform-Mediated Positive Feedback Loop. *Cancer Res* **77**, 3791-3801, doi:10.1158/0008-5472.CAN-16-2545 (2017).
- 2 Dharaneeswaran, H. *et al.* FOXO1-mediated activation of Akt plays a critical role in vascular homeostasis. *Circ Res* **115**, 238-251, doi:10.1161/CIRCRESAHA.115.303227 (2014).

Reviewers' comments:

Reviewer #1 (Remarks to the Author):

no issues with the revised manuscript

Reviewer #2 (Remarks to the Author):

In the revised version of the manuscript, "Receptors for insulin, not IGF-1, accelerate intimal hyperplasia by upregulation of hyaluronan synthase 2" the authors have been responsive to many of the previous concerns. This remains a provocative and interesting study and the in vivo data are intriguing. However, the data, in particular the signaling data and data supporting an insulin-Has2 link, are not robust to support the overall conclusions. More specific comments are below:

1. The primary in vivo data continue to compare SM22-Cre-IR KO mice to Myh11-Cre-IGF1R KO mice. As the authors generated Myh11-Cre-IR KO mice, this data should be presented in Figure 1, complete with proliferation data, and SM22-Cre-IR KO mouse data moved to Supplemental Figure 7.
2. Supplemental Figure 6 shows macrophage data, but this is not addressed in the text. Please include this as well as comparable macrophage data from Myh11-Cre-IR KO mice.
3. Figure 3: Blots are not consistent with quantitative data regarding pAkt (and pErk). This could potentially be due to varying degrees of IR or IGF1R depletion as SMCs were generated from IR KO and IGF1R KO mice (presumably SM22-Cre vs Myh11-Cre generated, but not addressed). Residual levels of each receptor could contribute to lack of robust signaling data. Authors demonstrated the ability to generate ~100% receptors knockout SMCs with the use of adeno-Cre. Perhaps using this system would improve the quality of these data where there would not be confounding issues with residual amounts of receptors.
4. Figure 5: panel A – it is quite possible that this panel is labeled incorrectly, but as presented there is some confusion and the blot is not consistent with the quantitative data. For instance, there is a lack of insulin pAkt response in WT SMCs that is inconsistent with previous figures (i.e. Figure 3). Why is there an increase in pAkt with insulin (or possibly IGF1) in DKO SMCs (lanes 4-6) if there are no receptors to activate Akt? Why is there an increase in pAkt in non-insulin-treated IR-ECD/IGF1R-ECD that is similar to insulin treated SMCs (compare lane 8 to 9)? If mislabeled, then insulin and IGF1 had comparable effects on pAkt in these cells which was not addressed. Panel E – There is a comparable induction of pAkt in double chimeric SMCs when compared to IR-ECD/IGF1R-ECD SMCs, but also a basal activation of Akt in the doubles. However, the text states that "insulin's effect on

pAkt in cells expressing IR-ECD/IGF1R-ECD and IGF1R-ICD/IR-ECD together were greatly decreased compared to cells which expressed only IR-ECD/IGF1R-ECD (lines 269-271).” This could also be interpreted as autoactivation in the absence of insulin in double chimeric overexpressing cells and should be addressed.

5. The link between insulin and Has2 is not convincing nor does it appear to be biologically significant.

6. Figure 7D: “The expression of Has2 mRNA was decreased in femoral artery of SMIRKO mice compared to WT control (lines 305-307).” This was not demonstrated in the graph.

7. Figure 7H: the graph shows significance between rapamycin-treated control vs insulin-treated SMCs, but not insulin-treated vehicle vs rapamycin treated SMCs. This should be addressed.

Reviewer #3 (Remarks to the Author):

My concerns were addressed by the new data and comments from authors.

Response to Reviewers' comments

Reviewer #1 (Remarks to the Author):

no issues with the revised manuscript

Reviewer #2 (Remarks to the Author):

In the revised version of the manuscript, "Receptors for insulin, not IGF-1, accelerate intimal hyperplasia by upregulation of hyaluronan synthase 2" the authors have been responsive to many of the previous concerns. This remains a provocative and interesting study and the in vivo data are intriguing. However, the data, in particular the signaling data and data supporting an insulin-Has2 link, are not robust to support the overall conclusions. More specific comments are below:

1. The primary in vivo data continue to compare SM22-Cre-IR KO mice to Myh11-Cre-IGF1R KO mice. As the authors generated Myh11-Cre-IR KO mice, this data should be presented in Figure 1, complete with proliferation data, and SM22-Cre-IR KO mouse data moved to Supplemental Figure 7.

We thank the reviewer for the suggestion which has been implemented. The femoral arteries of myh11-Cre-IR KO mice and control mice were subjected to wire injury. VSMC proliferation, as determined by BrdU and SM22a double staining at 7 days after injury was significantly decreased in Myh11IRKO mice compared to WT mice (1.98 ± 0.64 vs 8.8 ± 2.9 , $p < 0.05$). The new data are shown in Figure 1 E-F and described in line 135-137 of the Results section. VSMCs were also cultured from aortas of control or Myh11IRKO mice and cellular proliferation was determined by EdU incorporation. Insulin increased cellular proliferation by 2.7-fold in WT VSMCs, but only 1.5-fold in Myh11IRKO VSMCs. The new data are shown in Figure 1G and described from Line 137 to 139 of the Result section.

2. Supplemental Figure 6 shows macrophage data, but this is not addressed in the text. Please include this as well as comparable macrophage data from Myh11-Cre-IR KO mice.

We have added the following statement in Line 166-168 of Result stating "The macrophages in the intimal area of femoral artery of WT mice were not different compared to that of SMIGF1RKO mice (13 ± 4 vs 18 ± 8 , $p = 0.47$).

3. Figure 3: Blots are not consistent with quantitative data regarding pAkt (and pErk). This could potentially be due to varying degrees of IR or IGF1R depletion as SMCs were generated from IR KO and IGF1R KO mice (presumably SM22-Cre vs Myh11-Cre generated, but not addressed). Residual levels of each receptor could contribute to lack of robust signaling data. Authors demonstrated the ability to generate ~100% receptors knockout SMCs with the use of adeno-Cre.

Perhaps using this system would improve the quality of these data where there would not be confounding issues with residual amounts of receptors.

We thank the reviewer for the comments. VSMCs from IR flox/flox mice were infected with AdCre to knockdown IR. Insulin receptor was deficient in AdCre transfected VSMCs. Insulin induced Akt phosphorylation was increased by 6.4-fold in WT VSMCs, but only 3-fold in IR deficient VSMCs. VSMCs from IGF1R flox/flox mice were infected with AdCre to knockdown IGF1R. IGF-1 stimulated Akt was significantly decreased in IGF1R flox/flox VSMCs infected with AdCre. The data are shown in Supplemental Figure 10 and described from Line 234 to 243 of the Result section.

4. (A).Figure 5: panel A – it is quite possible that this panel is labeled incorrectly, but as presented there is some confusion and the blot is not consistent with the quantitative data.

We thank the reviewer for pointing out that an error was made in the labeling of this panel.. It has been corrected.

(B).For instance, there is a lack of insulin pAkt response in WT SMCs that is inconsistent with previous figures (i.e. Figure 3).

We apologize for the error in labeling. Insulin increased Akt phosphorylation in WT VSMCs.

(C).Why is there an increase in pAkt with insulin (or possibly IGF1) in DKO SMCs (lanes 4-6) if there are no receptors to activate Akt?

It is likely that there was still some insulin receptor and IGF-1 receptors in DKO cells. The stimulations of pAkt by insulin or IGF-1 were likely due to the activation of these residual receptors.

(D).Why is there an increase in pAkt in non-insulin-treated IR-ECD/IGF1R-ICD that is similar to insulin treated SMCs (compare lane 8 to 9)?

This is due to mislabeling which has been corrected. Line 8 is insulin treated SMCs and line 9 is IGF-1 treated SMCs.

(E). If mislabeled, then insulin and IGF1 had comparable effects on pAkt in these cells which was not addressed.

We thanked the reviewer for raising this interesting point. The p-Akt induced by insulin was slightly higher than that by IGF-1 in IR-ECD/IGF1R-ICD cells, although the difference was not significant. The cells were stimulated with 100 nM insulin or 100 nM IGF1. These data suggest that the extracellular domain of insulin receptor could be fully activated by a high dose of IGF-1, whereas the extracellular domain of IGF-1 receptor could only be partially activated by a high dose of insulin. This comment is added to Discussion (From Line 400 to 404).

(F). Panel E – There is a comparable induction of pAkt in double chimeric SMCs when compared to IR-ECD/IGF1R-ECD SMCs, but also a basal activation of Akt in the doubles. However, the text states that “insulin’s effect on pAkt in cells expressing IR-ECD/IGF1R-ECD and IGF1R-ICD/IR-ECD together were greatly decreased compared to cells which expressed only IR-ECD/IGF1R-ECD

(lines 269-271).” This could also be interpreted as autoactivation in the absence of insulin in double chimeric overexpressing cells and should be addressed.

We thank the reviewer for this comment. It is interesting that the basal activation of Akt in double chimeric SMCs is increased, which might be due to autoactivation of double chimeric receptors in the absence of insulin. Cultured smooth muscle cells are known to produce IGF-1. Some of the double chimeric receptors formed hybrid chimeric receptors (IR-ECDIGF1R-ECD/IGF1R-ICDIR-ICD). Another possibility might be that the hybrid chimeric receptor was more sensitive to the trace amount of IGF-1 which were produced by smooth muscle cells. This discussion is described from Line 404 to 408 in the Discussion Section.

5. The link between insulin and Has2 is not convincing nor does it appear to be biologically significant.

The important role of Has2 in VSMCs proliferation has been reported by another group¹. In this study, Has2 SiRNA decreased Has2 expression by 50% and significantly inhibited VSMCs proliferation¹. We agree that insulin’s effect on Has2 expression is mild. However, a 50% change at expression level could result in a significant change of enzymatic activation of Has2. We observed a 2-fold increase of hyaluronan in the femoral artery of SMIGF1RKO mice after wire injury which is clearly significant (Figure 7E-F). So we believe the changes in Has2 are biologically significant.

6. Figure 7D: “The expression of Has2 mRNA was decreased in femoral artery of SMIRKO mice compared to WT control (lines 305-307).” This was not demonstrated in the graph.

We revised this statement and the new statement is “The expression of Has2 mRNA has a tendency to be decreased in femoral artery of SMIRKO mice compared to WT control (lines 305-307), but the difference was not significant (p=0.4).”

7. Figure 7H: the graph shows significance between rapamycin-treated control vs insulin-treated SMCs, but not insulin-treated vehicle vs rapamycin treated SMCs. This should be addressed.

Insulin significantly increased cell proliferation in vehicle treated VSMCs. However, the insulin effects was decreased by 64% in rapamycin treated VSMCs compared to vehicle treated VSMCs.

Reviewer #3 (Remarks to the Author):

My concerns were addressed by the new data and comments from authors. Reviewer #2 (Remarks to the Author):

Reference

- 1 van den Boom, M. *et al.* Differential regulation of hyaluronic acid synthase isoforms in human saphenous vein smooth muscle cells: possible implications for vein graft stenosis. *Circ Res* **98**, 36-44, doi:10.1161/01.RES.0000199263.67107.c0 (2006).

Reviewers' comments:

Reviewer #2 (Remarks to the Author):

The authors have been mostly responsive to my previous concerns. There remain a few issues that should be addressed, as described below.

- A description of how vessel remodeling was measured/calculated should be included in the methods section. The graphs in figures 1&2 and sup. figure 4 show “intimal area (ratio of WT)” as a measure of intimal hyperplasia. This is an inappropriate measure. Intima/Media ratio is the gold standard for measuring intimal hyperplasia and the authors should include a measure of “percent stenosis” (Circulation. 1990 Dec;82(6):2190-200). This is especially important given the IGF1R KO mice show no differences in I/M ratio, which likely skews interpretation of these data. Importantly, the WT mice shown in figure 2 are less representative of the WT mice shown in figure 1 (and more similar to the IR KO mice).
- Scale bars should be included on all microscopy images.
- The previous comment was made and answered: “Supplemental Figure 6 shows macrophage data, but this is not addressed in the text. Please include this as well as comparable macrophage data from Myh11-Cre-IR KO mice. We have added the following statement in Line 166-168 of Result stating “The macrophages in the intimal area of femoral artery of WT mice were not different compared to that of SMIGF1RKO mice (13±4 vs 18±8, p=0.47).” However, the text reads “The macrophages and the collagen content (trichrome staining) in the intimal area of femoral artery of WT mice were not different compared to that of SMIGF1RKO mice (13±4 vs 18±8, p=0.47, Suppl, Figure 8).” The macrophage data is not included in this revised version.
- The authors should be very clear when describing the SMC cell systems’ they used (for each figure, detail if SMCs were directly derived from SM22-IR KO, Myh11-IR KO, Myh11-IGF1R KO mice, SMCs isolated from IR or EGF1R floxed mice and treated with adeno-Cre, or WT SMCs treated with IR- or IGF1r-specific siRNA).
- The data regarding overexpression of constitutively active Foxo1 are confusing and in opposition to what the authors propose. Phosphorylation of Foxo1 inhibits its activity and, as stated (line 318), “Foxo1 could bind to the promoter of Has2 and inhibit the transcription of Has2”. Therefore

decreased expression of Has2 would be expected with constitutively active Foxo1. In contrast, the authors report increased Has2 expression with caFoxo1. This should be addressed.

Response to Reviewers' comments

The authors have been mostly responsive to my previous concerns. There remain a few issues that should be addressed, as described below.

1. A description of how vessel remodeling was measured/calculated should be included in the methods section. The graphs in figures 1&2 and sup. figure 4 show “intimal area (ratio of WT)” as a measure of intimal hyperplasia. This is an inappropriate measure. Intima/Media ratio is the gold standard for measuring intimal hyperplasia and the authors should include a measure of “percent stenosis” (Circulation. 1990 Dec;82(6):2190-200). This is especially important given the IGF1R KO mice show no differences in I/M ratio, which likely skews interpretation of these data. Importantly, the WT mice shown in figure 2 are less representative of the WT mice shown in figure 1 (and more similar to the IR KO mice).

We thank the reviewer for pointing out that I/M ratio is the gold standard for measuring intimal hyperplasia, which we have shown in Figure 1C to be decreased in Myh11IRKO mice compared to WT mice. This result was different from Figure 2C which showed that I/M ratio did not differ between SMIGF1RKO mice and WT mice. We have restated our conclusion from “Our findings using specific deletions of IR and IGF1R in VSMC provide the surprising and clear result that the loss of IGF1R does not protect and actually enhances intimal hyperplasia, whereas deletion of IR decreases VSMC proliferation and intimal hyperplasia following intimal injury” to “Our findings using specific deletions of IR and IGF1R in VSMC provide the surprising and clear result that the loss of IGF1R does not protect against intimal hyperplasia, whereas deletion of IR decreases VSMC proliferation and intimal hyperplasia following intimal injury.” (Line 333-336). As for the additional analysis to assess “percent stenosis”, most recent publications on this subject did not use percent stenosis to quantify intimal hyperplasia¹⁻⁴. “Percent stenosis” is a measurement for restenosis which is different from intimal hyperplasia. Restenosis is more complicated to assess compared to intimal hyperplasia and is also regulated by artery tone and elastic

fiber degradation. We will investigate the effects of IGF1 on artery tone and elastic fiber degradation in future studies.

The intimal hyperplasia induced by wire injury was affected by multiple factors, including metabolic and environment factors. We used littermate control mice in femoral artery wire injury experiments. The WT mice used in Figure 1A were IR flox/flox mice, whereas the WT mice used in Figure 2A were IGF1R flox/flox mice. So the WT mice used in Figure 1A and Figure 2A were not exactly same. The difference in the WT mice of Figure 1A and Figure 2A could be due to variation. This is the reason that we used littermate controls which is much more time consuming to do.

2. Scale bars should be included on all microscopy images.

Scale bars have been added to all microscopy images.

3. The previous comment was made and answered: “Supplemental Figure 6 shows macrophage data, but this is not addressed in the text. Please include this as well as comparable macrophage data from Myh11-Cre-IR KO mice. We have added the following statement in Line 166-168 of Result stating “The macrophages in the intimal area of femoral artery of WT mice were not different compared to that of SMIGF1RKO mice (13±4 vs 18±8, p=0.47).” However, the text reads “The macrophages and the collagen content (trichrome staining) in the intimal area of femoral artery of WT mice were not different compared to that of SMIGF1RKO mice (13±4 vs 18±8, p=0.47, Suppl, Figure 8).” The macrophage data is not included in this revised version.

The macrophage data has been added to Supplemental Figure 8A.

4. The authors should be very clear when describing the SMC cell systems’ they used (for each figure, detail if SMCs were directly derived from SM22-IR KO, Myh11-IR KO, Myh11-IGF1R KO

mice, SMCs isolated from IR or EGF1R floxed mice and treated with adeno-Cre, or WT SMCs treated with IR- or IFG1r-specific siRNA).

We followed the reviewer's suggestion and described the origin of VSMCs in Figure legends.

5. The data regarding overexpression of constitutively active Foxo1 are confusing and in opposition to what the authors propose. Phosphorylation of Foxo1 inhibits its activity and, as stated (line 318), “Foxo1 could bind to the promoter of Has2 and inhibit the transcription of Has2”. Therefore decreased expression of Has2 would be expected with constitutively active Foxo1. In contrast, the authors report increased Has2 expression with caFoxo1. This should be addressed.

We thank the reviewer for the question regarding increased basal Has2 expression with CA-FoxO1. We agree that this description could be confusing. The finding is not opposite to our proposed theory since it is known that activation of Akt could increase the activity of mTORC, which in turn can increase the expression of Has2^{5,6}. We showed that insulin upregulated Has2 expression through p-Akt. It is known that overexpression of CA-FoxO1 increases Akt and mTORC activation⁷, so overexpression of CA-FoxO1 elevates the basal level of Has2 through activating mTORC. This is the reason we performed the study using rapamycin to understand the role of mTORC in the regulation of Has2 expression. The regulation of Has2 is complicated. Activation of Akt regulates Has2 expression through both insulin- and non-insulin dependent pathways. CA-FoxO1 inhibits insulin action, but it can also increase Has2 expression through non-insulin dependent, mTORC dependent, mechanisms. We hope this explanation clarifies the data of CA-FoxO1 and Has2 expression in SMC.

1 Guo, X. *et al.* Dedicator of cytokinesis 2, a novel regulator for smooth muscle phenotypic modulation and vascular remodeling. *Circ Res* **116**, e71-80, doi:10.1161/CIRCRESAHA.116.305863 (2015).

2 Liu, R. *et al.* Ten-eleven translocation-2 (TET2) is a master regulator of smooth muscle cell plasticity. *Circulation* **128**, 2047-2057, doi:10.1161/CIRCULATIONAHA.113.002887 (2013).

- 3 Pavic, G. *et al.* Thrombin receptor protease-activated receptor 4 is a key regulator of exaggerated intimal thickening in diabetes mellitus. *Circulation* **130**, 1700-1711, doi:10.1161/CIRCULATIONAHA.113.007590 (2014).
- 4 Xie, N. *et al.* SRSF1 promotes vascular smooth muscle cell proliferation through a Delta133p53/EGR1/KLF5 pathway. *Nat Commun* **8**, 16016, doi:10.1038/ncomms16016 (2017).
- 5 Kultti, A. *et al.* Methyl-beta-cyclodextrin suppresses hyaluronan synthesis by down-regulation of hyaluronan synthase 2 through inhibition of Akt. *J Biol Chem* **285**, 22901-22910, doi:10.1074/jbc.M109.088435 (2010).
- 6 Qin, J., Berdyshev, E., Poirer, C., Schwartz, N. B. & Dawson, G. Neutral sphingomyelinase 2 deficiency increases hyaluronan synthesis by up-regulation of Hyaluronan synthase 2 through decreased ceramide production and activation of Akt. *J Biol Chem* **287**, 13620-13632, doi:10.1074/jbc.M111.304857 (2012).
- 7 Dharaneeswaran, H. *et al.* FOXO1-mediated activation of Akt plays a critical role in vascular homeostasis. *Circ Res* **115**, 238-251, doi:10.1161/CIRCRESAHA.115.303227 (2014).

REVIEWERS' COMMENTS:

Reviewer #2 (Remarks to the Author):

I have no further comments on this paper. The authors adequately addressed my previous concerns.